# Suitability Analysis of Ski Areas in China: An Integrated Study Based on Natural and Socioeconomic Conditions

Jie Deng[1,2,5], Tao Che[1,2], Cunde Xiao[3], Shijin Wang[4], Liyun Dai[1], and Akynbekkyzy Meerzhan[1,5,6]

[1]Heihe Remote Sensing Experimental Research Station, Key Laboratory of Remote Sensing of Gansu Province, Northwest Institute of Eco-Environment and Resources, Chinese Academy of Sciences, Lanzhou, 730000, China
[2]Center for Excellence in Tibetan Plateau Earth Sciences, Chinese Academy of Sciences, Beijing, 100101, China
[3]State Key Laboratory of Earth Surface Processes and Resource Ecology, Faculty of Geographical Science, Beijing Normal University, Beijing, 100000, China
[4]State Key Laboratory of Cryospheric Science, Northwest Institute of Eco-Environment and Resources, Chinese Academy of Sciences, Lanzhou, 730000, China
[5]University of Chinese Academy of Sciences, Beijing, 100049, China
[6]Jiangsu Center of Collaborative Innovation in Geographical Information Resource Development and Application, Nanjing, 210000, China

*Corresponding to*: Tao Che (chetao@lzb.ac.cn)

**Abstract.** The successful bid for the 2022 Winter Olympics (Beijing 2022, officially known as the XXIV Olympic Winter Games) has greatly stimulated Chinese enthusiasm towards winter sports participation. Consequently, the Chinese ski industry is rapidly booming due to enormous market demand and government support. However, investing in ski areas in unreasonable locations will cause problems from an economic perspective (in terms of operation and management) as well as geographical concerns (such as environmental degradation). Therefore, evaluating the suitability of a ski area based on scientific metrics has become a prerequisite for the sustainable development of the ski industry. In this study, we evaluate the locational suitability of ski areas in China by integrating their natural and socioeconomic conditions using a linear weighted method based on geographic information system (GIS) spatial analysis combined with remote sensing, online and field survey data. The key indexes for evaluating natural suitability include snow cover, air temperature, topographic conditions, water resources, and vegetation, whereas socioeconomic suitability is evaluated based on economic conditions, accessibility of transportation, distance to a tourist attraction, and distance to a city. As such, metrics ranging from 0 to 1 considering both natural and socioeconomic conditions are used to define a suitability threshold for each candidate region for ski area development. A ski area is considered to be a dismal prospect when the locational integrated index is less than 0.5. The results show that 84% of existing ski areas are located in areas with an integrated index greater than 0.5. Finally, corresponding development strategies for decision-makers are proposed based on the multicriteria metrics, which will be extended to incorporate potential influences from future climate change and socioeconomic development. However, the snowmaking model with local data should to be used to further analyze the suitability for a specific ski area.

# 1 Introduction

Ski tourism, as a major component of winter sports and tourism, creates great opportunities for business and promotes regional economic development (Eadington and Redman, 1991). In many countries, ski tourism has brought in more revenue than traditional agricultural and industrial practices (Lasanta, Laguna, and Vicente-Serrano, 2007). Especially in mountainous regions, the development of ski tourism has opened new doors for employment, which has boosted the local economy and led to the recovery of the local population (Silberman and Rees, 2010).

It is well known that environmental sustainability is often sacrificed during the rapid development of the economy, which, in turn, hampers future business opportunities. The operation of ski areas is generally accompanied by environmental degradation, such as deforestation, vegetation destruction and soil erosion (Burt, 2012). The construction and maintenance of ski slopes have been found to cause disturbances in mountainous regions, which have significant impacts on the ecosystem and the environment (Burt and Rice, 2009). With the expansion of such infrastructure into nature, this kind of human-induced disturbance compels animals to leave their original habitat, leading to habitat loss and fragmentation (Sato et al., 2014; Brambilla et al., 2016). The enlargement of ski runs and the use of artificial snow are particularly prone to causing changes in plant species and declines in biodiversity (Wipf et al., 2005; Delgado et al., 2007), which may further increase surface runoff and soil erosion, and ultimately lead to land degradation (Ristić et al., 2012). Furthermore, constrained sediments and chloride have been found in streams near ski areas (Wemple et al., 2007). Extensive studies of lakes in northern Finland have shown that ski area operations have resulted in water contamination, such as eutrophication (Tolvanen and Kangas, 2016).

For the sustainable development of ski areas, the impact of climate change has to be considered because 1) snow is the most important resource that supplies the ski area and 2) locally increased temperature may shorten the duration of the ski season (Steiger and Abegg, 2018; Gilaberte-Búrdalo et al., 2017). Previous studies have established national or regional climate-driven snow models under different climate change scenarios to investigate how future climate change will affect snow conditions (Hennessy et al., 2008; Rutty et al., 2017; Verfaillie et al., 2018). These studies led to the conclusions that global warming will lead to a decline in the snowpack and a shortening of the length of the ski season. In addition, winter sports are vulnerable to climate variability, particularly at lower elevation and in mid-latitude ski areas, because poor snow conditions are shown to result in a decline in the number of tourists (Scott, 2017). Snowmaking is usually preferred as a supplementary strategy for individual ski areas in response to worsening natural snow conditions (Hennessy et al., 2008; Pons-pons et al., 2012). Machine-made snow can remarkably improve the reliability of snow conditions in both current and future climate scenarios, especially in regions with less reliable snow conditions and short ski seasons (Steiger, 2010; Hanzer, Marke, and Strasser, 2014; Damm, Koeberl, and Prettenthaler, 2014; Damm et al. 2017). In addition, snowmaking results in increased water and energy consumption, which contradicts the well-acknowledged consensus regarding the reduction in greenhouse gas emissions to mitigate the effects of climate change (Steiger and Stötter, 2013). The efficiency of converting water into snow during snowmaking is less than 80% because of the influence of meteorological conditions (Spandre et al., 2017). For example, at least 2 million liters of water are needed to cover 1 ha with a 30 cm thick snowpack. In terms of electric

consumption, the operation of a snow gun currently requires 2 to 3 kWh to produce 1 m$^3$ of snowpack. Therefore, intense snowmaking would pose critical economic and environmental threats to ski areas, which might lead to their permanent closure (Tervo-kankare, Kaján, and Saarinen, 2017). In summary, the locations of ski areas are critical since inappropriate choices may contaminate, degrade or even damage the environment (Tsuyuzaki, 1994). Evaluating ski areas locations based on scientific metrics is a prerequisite to help reduce the inevitable consequences caused by construction and operation (Spulerova et al., 2016; Cai et al., 2019).

Recently, in major skiing countries, the number of ski resorts has become relatively saturated or declined (such as in Japan, Vanat, 2019). In contrast, the successful bid for the 2022 Winter Olympics (Beijing 2022, officially known as the XXIV Olympic Winter Games) has motivated the general public in China to participate in ice and snow sports. As a competitive sport with potential business opportunities, skiing is strongly supported by the Chinese government, which has created a good atmosphere for the associated promotion and training. With a series of initiatives and future plans proposed by the Chinese government, it is expected that 300 million citizens will invest in winter sports activities before 2022. In 2010, there were only 270 ski areas in China. However, driven by enormous market demands and government support, the number of ski areas has been rapidly increasing in recent years. Thirty-nine new ski areas were opened in 2018, bringing the total number of ski areas in China to 742 (Vanat, 2019), and this is only the start of the upcoming extensive growth in ski tourism. According to the strategic plan proposed by the China National Tourism Administration, the number of ski areas is expected to reach 800 by 2022. This booming industry raises the following question: how can we mitigate the environmental problems and avoid unnecessary economic losses caused by the unreasonable locations of ski areas?

The operation of ski areas mainly relies on a suitable natural environment and local socioeconomic conditions (Morey, 1985). On the one hand, ski areas are highly controlled by natural conditions, such as terrain and climate conditions (Morey, 1984; Gilaberte-Búrdalo et al., 2017). On the other hand, local socioeconomic conditions are key indicators supporting the operation of ski areas because the market for most ski areas is mainly composed of local tourists (Silberman and Rees, 2010). Thus, both natural and socioeconomic conditions should be considered to determine the locations of ski areas (Fukushima et al., 2002). Despite the pivotal role that ski areas have played in the social economy and environment, only a few studies have focused on locational suitability during the development of ski areas (e.g., Geneletti, 2008; Silberman and Rees, 2010; Dezsi et al., 2015; Cai, Di, and Liu, 2019). For example, Silberman and Rees (2010) evaluated alternative but currently undeveloped sites based on the standard measures of existing areas, and the conclusions of that study could be applied for decision-making in the industry. Moreover, previous studies are mostly limited to small scales (i.e., local or regional scales). In this study, geographic information system (GIS) spatial analysis is combined with remote sensing, online and field survey data to employ a linear weighted method to evaluate the suitability of ski area location in China based on both natural and socioeconomic conditions. This work can provide scientific metrics for decision-makers to avoid unreasonable economic investments and environmental problems and thus promote the sustainable development of Chinese ski tourism.

This paper is organized as follows: in Section 2, we describe the data and methods. Section 3 presents the results. Section 4 validates the method, evaluates the locational suitability of existing ski areas, proposes suggestions for a development strategy, and discusses the limitations of this work. The final section contains a brief conclusion and discusses future work.

## 2 Data and Methodology

### 2.1 Data collection

In this study, the suitability of ski area locations was evaluated based on two components: the natural conditions (from the supply perspective) and the socioeconomic conditions (from the demand perspective). Regions where the elevation is higher than 4000 m are unsuitable for skiing due to the lack of oxygen, as are the areas where the maximum air temperature is higher than 10 °C for more than 90 days during the ski season (November 1 to March 31) (Scott, McBoyle, and Mills, 2003). Areas with high elevations were mainly distributed in western China, including most of the Tibetan Plateau, while areas with high temperatures were mainly distributed in southern China and the Sichuan Basin (see Fig. 1a). All factors were analyzed using GIS spatial analysis based on data from multiple sources with a spatial resolution of 1 km combined with online and field survey data. Table 1 summarizes all of the data used in this study. Normalization processing was implemented for all natural and socioeconomic indexes to make the data dimensionless and mutually comparable (Geneletti, 2008). This work was conducted for the entirety of China, except for Taiwan, Macao and Hong Kong, due to the lack of data.

### 2.1.1 Natural conditions

Variables serving as indexes of natural conditions that could impact the development of ski areas were used, including snow cover, air temperature, topographic conditions, water resources and vegetation, as shown in Fig. 1.

**Snow cover**

Natural snow cover is a crucial resource for ski areas. Skiers may cancel trips when there are poor snow conditions (Scott, McBoyle, and Mills, 2003; Steiger and Abegg, 2018). Tervo (2008) analyzed the viability of nature-based winter tourism enterprises and declared that 90–120 skiable days are adequate for making a profit. In fact, a ski area is profitable if the snow reliability period is greater than 100 days per season, which is known as the 100-day rule and is the most common indicator of snow reliability (Steiger, 2012). Additionally, Scott, McBoyle, and Mills (2003) defined a skiable day as a day with a snow depth greater than 30 cm on ski runs. However, the snow depth in China is much lower than that in North America and Europe, and the areas with natural snow depths greater than 30 cm are extremely rare (Mudryk et al., 2015). Therefore, we did not use the indicator of the number of days with a natural snow depth greater than 30 cm. As a supplement for the snow depth on ski runs, the average snow depth (SD) during a ski season was considered in this study. In addition, the number of snow cover days (SCD), which is total number of days (can be discontinuous) with snow cover in an area during the ski season, was considered as an indicator of climate suitability (temperature and precipitation) and a measure of the aesthetic value of a site given that many ski areas have snow cover on only the ski runs.

With a spatial resolution of 500 m, version 4 of the moderate-resolution imaging spectroradiometer (MODIS) snow cover products MOD10A1 and MYD10A1 were used in this study to obtain SCD, and these products were downloaded from the National Snow Ice Data Center (NSIDC, https://nsidc.org/). The normalized difference snow index (NDSI) was adopted to distinguish snow in the MODIS snow cover images. The NDSI was calculated using the fourth and sixth bands of MODIS, and a pixel was defined as snow if the NDSI value was greater than 0.4. Further details on the MODIS snow cover products can be found in Hall et al. (2002). In this study, the MODIS snow cover images were resampled to 1 km. The SD products were downloaded from the Cold and Arid Regions Sciences Data Center at Lanzhou (http://westdc.westgis.ac.cn). The SD products derived from passive microwave remote-sensing data in China were developed by Che et al. (2008), with a spatial resolution of 25 km. Based on the MODIS cloud removal algorithm and downscaling algorithm developed by Huang et al. (2016), the daily cloud-free snow cover data and 1 km daily SD data for the 2010-2014 ski seasons were produced, and then the average SCD and SD from 2010 to 2014 were calculated. An SCD larger than 100 days and SD greater than 30 cm were taken as the optimal conditions for normalization. The snow cover index was defined as the sum of the normalized average SD and SCD, each assigned a weight of 0.5 (Fig. 1b). This index was for natural snow only, and machine-made snow was taken into account by the index of air temperature.

**Air temperature**

Low air temperature is a prerequisite for skiing. Scott, McBoyle, and Mills (2003) noted that ski areas should be closed if the maximum temperature is greater than 10 ℃ for 2 consecutive days accompanied by liquid precipitation. Additionally, skiers may also choose not to ski at extremely low temperatures (below ~-25 ℃) (Tervo, 2008; Rutty and Andrey, 2014). Because snowmaking has been used as an adaptive strategy to compensate for the decrease in snow reliability worldwide, temperatures between -2 ℃ and -5 ℃ were taken as optimal conditions in regard to efficient snowmaking (Tervo, 2008).

Based on the daily air temperature generated from more than 2400 meteorological stations in China, a gridded daily observation dataset with a spatial resolution of 25 km was developed by Wu and Gao (2013). The gridded daily observation dataset includes two variables: daily mean air temperature and daily maximum air temperature. The daily mean air temperature was the average value of the dry-bulb temperature at 02:00, 8:00, 14:00 and 20:00 (local time). The daily maximum air temperature was measured using a maximum thermometer. In this study, using an extrapolation method with a lapse rate of 0.65 ℃ per 100 m of elevation (Steiger and Stötter, 2013), the temperature dataset was resampled to 1 km spatial resolution for the 2010-2014 ski seasons. The daily maximum air temperature dataset was used to obtain the high-temperature regions that are unsuitable for ski area development. In the southern China, many small ski areas can be profitable during a ski season with 60 skiable days. Therefore, the high-temperature region was defined as the area with more than 90 noncontinuous days with maximum air temperatures greater than 10 ℃ during a ski season (151 days). Following previous studies (Scott, McBoyle, and Mills, 2003; Tervo, 2008; Rutty and Andrey, 2014), the daily mean air temperature was reclassified into 11 regimes with corresponding scores (Table 2), and the air temperature index was the mean score for the 2010-2014 ski seasons (Fig. 1c). The 11 temperature regimes and their corresponding scores were designed as a trade-off between the cold temperatures needed to preserve the snowpack and to produce machine-made snow and the warm temperatures needed by skiers.

**Topographic conditions**

Topographic conditions are essential for identifying suitable locations for ski areas. Ski runs require a certain topographic slope (Dezsi et al., 2015). In 2014, the China National Tourism Administration announced the Rank Division for the Quality of Mountain Ski Resorts, which requires that the average slope gradient of advanced slopes should not be less than 20° and that the average slope gradient of intermediate slopes should be greater than 15°. Nevertheless, the conditions are dangerous for most tourists when the slope is too steep (slope > 30°). Unfortunately, information loss is inevitable when using the slope values extracted from remote sensing images during averaging. Accordingly, slope gradients of 10° to 20° were considered to be the best topographic conditions.

Slope gradient data were derived from the Shuttle Radar Topography Mission (SRTM) digital elevation model (DEM) (V004) with a spatial resolution of 90 m (National Map Seamless Data Distribution Systems, http://seamless.usgs.gov). All pixels with values greater than 30° were assigned a value of 30°. The topographic conditions index was produced by a normalized slope gradient (Fig. 1d). The normalized value was 1 if the slope gradient was between 10° and 20°. The greater the absolute difference is from this interval, the lower the normalization value. When the slope gradient is 0° or larger than 30°, the normalized value is 0.

**Water resources**

To reduce the vulnerability of ski areas caused by climate change, snowmaking has been increasingly adopted as an adaptive strategy to compensate for the scarcity of natural snow (Hennessy et al., 2008; Pons-pons et al., 2012). In fact, snowmaking requires large amounts of water. The high water consumption of snowmaking not only increases the operating costs of ski areas but also poses stress on the local water resources, especially in arid regions (Duglio and Beltramo, 2016; Wilson et al., 2018). Consequently, an area with abundant water resources is more suitable for ski area development. Data on the spatial distribution of rivers and lakes were acquired from the Data Center for Resources and Environmental Sciences (RESDC, http://www.resdc.cn) of the Chinese Academy of Sciences (CAS). The water resources index is the sum of the cost distance to a river and the cost distance to a lake, each normalized to a range of [0-1] and given a weight of 0.5 (Fig. 1e). The higher the cost distance is, the lower the normalized value. The cost distance to a river (lake) was estimated using the cost distance method, which calculated the least cumulative cost distance for each pixel to the nearest river (lake) over a cost surface. The cost surface was a raster of slope gradients, which defined the cost to move planimetrically through each pixel. The value of the slope gradient at each pixel location represented the cost per unit distance to move through the pixel.

**Vegetation**

Vegetation contributes to creating an ideal environment and is an important indicator for evaluating the attractiveness of a destination. Vegetation, especially trees, can define the edges of ski runs and prevent skiers from being hit by strong winds (e.g., skiable days include the requirement of winds less than 6.5 m s$^{-1}$; Crowe, McKay, and Baker, 1973). The vegetation index for 2015 was derived based on a dataset of the spatial distribution of the annual normalized difference vegetation index (NDVI) covering the Chinese region (Fig. 1f). NDVI greater than 0.6 was the optimal value for normalization. The lower the

NDVI is, the lower the normalized value. With a spatial resolution of 1 km, this dataset integrated SPOT NDVI data and MODIS vegetation data, which were sourced from the RESDC (http://www.resdc.cn).

### 2.1.2 Socioeconomic conditions

Socioeconomic conditions highlight a favorable level of economic development and infrastructure conditions by integrating four indexes: economic conditions, distance to a city, accessibility of transportation, and distance to a tourist attraction. All indexes were reclassified into 10 regimes by geometrical interval classification (see Fig 2). The classification scheme creates 10 geometrical intervals by minimizing the square sum of elements per class. This procedure ensured that each class range had approximately the same number of values for each class and that the change between intervals was fairly consistent.

**Economic conditions**

Skiing is a luxury activity for skiers; therefore, the development of ski areas should consider the balance between investment and revenue. Economic census data are sometimes difficult to acquire over a large scale, and currently, reliable economic data in China are mainly reported at the provincial level (Zhao et al., 2017). With the development of remote sensing technology, nighttime light images have become an efficient means of mapping national economic activities (Rybnikova and Portnov, 2015). Visible infrared imaging radiometer suite (VIIRS) day/night band (DNB) nighttime light data are considered a new generation of nighttime light imagery. Composite images of the annual average radiance were produced by the Earth Observations Group (EOG) at the National Oceanic and Atmospheric Administration's National Geophysical Data Center (NOAA/NGDC) using VIIRS DNB nighttime data; the spatial resolution is 500 m. Prior to averaging, the DNB data were filtered to exclude data impacted by stray light, lightning, lunar illumination, and cloud cover. The data are available at https://www.ngdc.noaa.gov/eog/viirs/download_dnb_composites.html.

Many studies report that there is a strong correlation between an administrative unit's total gross domestic product (GDP) and the digital number (DN) value of the corresponding pixel in nighttime light imagery (Doll, Muller, and Elvidge, 2000; Zhao et al., 2011). In this study, without a direct estimation of the GDP values, nighttime light pixel values were used as a surrogate index to measure the economic development level in different areas. The economic conditions index was generated by kernel density analysis based on the composite images of the annual average radiance for 2015 (Fig. 2a). The higher the nuclear density value is, the higher the normalized value. All pixels greater than 0 in the composite images of annual average radiance were converted to point features. The kernel density method was used to calculate the magnitude per unit area from the point feature using a kernel function to fit a smoothly tapered surface to each point.

**Distance to a city**

The customer plays a major role in the ski tourism industry. Except for a few ski areas in China that can attract both national and international visitors, the skiers in most ski areas are local visitors. The farther a ski area is from a major metropolitan area, the less likely it is to draw a large crowd (Fukushima et al., 2002). As analyzed using the distance decay theory, the number of visitors declines exponentially as the distance from a source increases, although the impact of distance on visitors is not a deterministic construct in its own right (Mckercher, 2018).

Here, the index of the distance to a city was used to represent the rule of the distance decay theory (the farther away from a city, the fewer visitors the ski area can attract). The cities used for this study included provincial capitals, prefecture-level cities and county-level cities with airports. According to the distribution of public-use airports, these cities were divided into three categories: provincial capitals, cities with an airport, and cities without an airport. The distance to a city was estimated by calculating the least cost distance between any point in the study area and the nearest city. The distance to a city index was defined as the sum of the distance to a provincial capital, the distance to a city with an airport and the distance to a city without an airport, each normalized to a range of [0-1] and assigned weights of 0.5, 0.3 and 0.2, respectively (Fig. 2b). Information on the location of cities was taken from the National Geomatics Center of China (NGCC, http://www.ngcc.cn/), and the list of public-use airports in China was sourced from the Civil Aviation Administration of China (CAAC, http://www.caac.gov.cn/).

**Accessibility of transportation**

A transportation network is the foundation of accessibility. Kaenzig, Rebetez, and Serquet (2016) surveyed a former ski slope, and their results showed that easy access to a summit was the key factor in renewed attraction for visitors. Convenient transportation, which provides easy access to a ski destination, will bring more visitors. In this study, the accessibility of transportation index was acquired by calculating the least cost distance from any point to the nearest road (Fig. 2c). The road network was obtained from OpenStreetMap (https://www.openstreetmap.org/#map), which was assembled using a manual survey, global positioning system (GPS) devices, aerial photography, and other free sources.

**Distance to a tourist attraction**

Many enterprises are aware of the potential benefits of regional cooperation (Buhalis, 2000; Young, 2002). To reduce costs, maximize tourist attraction, and improve enterprise competitiveness, many ski areas, especially small ski areas, should be located within tourist attractions with necessary facilities, such as hotels, restaurants and shops (Lasanta, Laguna, and Vicente-Serrano, 2007; Dezsi et al., 2015; Spulerova et al., 2016). Additionally, visitors prefer to enjoy beautiful views while skiing. The distance to a tourist attractions index was determined by measuring the least cost distance from any point in the study area to the nearest tourist attraction (Fig. 2d). The tourist attractions were manually extracted from an online map (https://maps.baidu.com).

**2.2 Ski area information and field survey**

Although there were indeed 742 ski areas in China according to the International Report on Snow and Mountain Tourism in 2018 (Vanat, 2019), only 598 ski areas' information was manually collected from an online map (https://maps.baidu.com), because limited information was available online for lower-class ski areas. The collected ski areas were divided into three categories:

(1) 116 ski areas before 2012 as sample data: It is often believed that successful and suitable ski areas could operate for long periods of time. We then considered ski areas that had been in operation for more than 6 years as successful and suitable ski areas. As such, the location information for 116 operating ski areas established before 2012 (mainly distributed in northern,

northeastern and northwestern China, Fig. 7) was categorized as suitable sample data, which were used to calculate the weight coefficients for the different indexes.

(2) 35 ski areas during field surveys in 2018 for validation: To validate the results, field surveys on ski areas were conducted during the 2018 ski season over 13 provinces, spatially representing different natural and socioeconomic conditions (see Fig. 7). The managers or staff of 35 ski areas were interviewed with a questionnaire and through conversations, mainly involving location information, ski season length, the quality of the ski area, its operating state, and its market competition situation. The questions concerning the quality of the ski area were also designed to collect the characteristics of the ski area, including the number and slope of ski runs, equipment, facilities, and accessibility of transportation. According to the Rank Division for the Quality of Mountain Ski Resorts released by the China National Tourism Administration in 2014, the quality of ski areas was evaluated by five grades from 1S to 5S, representing the lowest- to the highest-grade standard. This procedure involved comprehensive assessment integrating 10 aspects, such as equipment, facilities, accessibility of transportation, and the reception scale. According to the feedback from our field surveys, ski areas were roughly ranked into different grades, except for the 8 ski areas whose grades were already defined by the Xinjiang Government Tourist Office.

(3) 447 ski areas for evaluation: Additionally, the locational suitability of the 447 ski areas established after 2012 was evaluated using the aggregation method.

## 2.3 Methodology

In this study, the linear weighting method was used for synthetic evaluation. The natural suitability for ski area development ($NS$) is expressed as

$$NS = \sum_{i=1}^{i} D_i NC_i \tag{1}$$

where $i$ is the number of natural condition indexes ($i = 1, 2,…, 5$), $D_i$ is the weighting factor that represents the importance of the natural condition indexes, and $NC_i$ represents the natural index.

The socioeconomic suitability for ski area development ($SS$) is expressed as

$$SS = \sum_{j=1}^{j} T_j SC_j \tag{2}$$

where $j$ is the number of the socioeconomic condition indexes ($j = 1, 2,…, 4$), $T_j$ is the weighting factor that represents the importance of the socioeconomic indexes, and $SC_j$ is the socioeconomic index.

The integrated suitability for ski area development ($IS$) is expressed as

$$IS = \sum_{f=1}^{f} H_f C_f \tag{3}$$

where $f$ is the number of indexes ($f = 1, 2$), $H_f$ is the weight coefficient, $C_f$ is the index of $NS$ or the index of $SS$.

The weight coefficients in Eqs. (1-3) ($D_i$, $T_j$ and $H_f$) are defined using the entropy weight method introduced as follows. Finally, $NS$, $SS$ and $IS$ are rescaled within the range of 0-1.

The 116 existing ski areas established before 2012 were used as samples to compute the weight coefficients by extracting all index values for the ski area location. The weight coefficients were calculated by an objective method based on the theory of

information entropy, which has been widely employed for the determination of weights of evaluating indicators (Bian et al., 2018; Bednarik et al., 2010; Srdjevic, Medeiros, and Faria, 2004; Vranešević et al., 2016). The concept of information entropy originally came from thermodynamics and indicates the extent of disorder in the system status (Bednarik et al., 2010). Generally, if the dispersion of data is high, the value of information entropy is low, which means that more information will

be provided. Correspondingly, the greater the influence of the index on the evaluation, the higher its weight (Tang, 2015). Therefore, the entropy weight method can be used to calculate the objective weights of the index system and avoid bias caused by subjectivity to a certain extent (Pourghasemi, Mohammady, and Pradhan, 2012).

It is supposed that there are $a$ optional schemes, each with $b$ evaluating indexes. The data matrix is established as follows:

$$X = \begin{bmatrix} x_{11} & \cdots & x_{1a} \\ \vdots & \ddots & \vdots \\ x_{b1} & \cdots & x_{ba} \end{bmatrix}_{a \times b} \tag{4}$$

where $x_{nm}$ is the value of the $m^{th}$ index in the $n^{th}$ scheme, $n = 1, 2,..., a$; $m = 1, 2,..., b$. $a$ is the number of ski areas established before 2012 ($a = 116$), $b$ is the number of indexes in Eqs. (1-3) ($i$, $j$ and $f$).

it is noted that the indexes are not measured on the same scale, and they have different dimensions. If the analysis is performed directly with the original index values, the difference in different quantity grades in $X$ may produce inaccurate results in the decision-making process. Therefore, normalization ($Z_{nm}$) is performed to make the indexes dimensionless and mutually

comparable, which is expressed as

$$Z_{nm} = \frac{x_{m,max} - x_{nm}}{x_{m,max} - x_{m,min}} \tag{5}$$

where

$$x_{m,max} = max\{x_{1m}, \ldots x_{am}\} \tag{6}$$

$$x_{m,min} = min\{x_{1m}, \ldots x_{am}\} \tag{7}$$

The information entropy $e_m$ of the $m^{th}$ index is defined as

$$e_m = -k \sum_{n=1}^{a} P_{nm} \ln(P_{nm}) \tag{8}$$

where

$$k = \frac{1}{\ln a} \tag{9}$$

$$P_{nm} = \frac{Z_{nm}}{\sum_{n=1}^{a} Z_{nm}} \tag{10}$$

if $P_{nm} = 0$, $P_{nm} \ln(P_{nm}) = 0$.

The entropy weight $w_m$ of the $m^{th}$ index is expressed as

$$w_m = \frac{1 - e_m}{\sum_{m=1}^{b} (1 - e_m)} \tag{11}$$

The calculation results of the weight coefficients for the evaluation indexes are shown in Table 3. The results show that the total weight of natural suitability (0.52) is higher than that of socioeconomic suitability (0.48), which indicates that natural

conditions have a greater impact on the development of ski areas than socioeconomic conditions.

## 3 Results

This section presents the evaluation results of the locational suitability for ski area development in China. The first part of the analysis focuses on the natural conditions for ski area development. Natural suitability was evaluated based on the linear weighting of five natural indexes: snow cover, air temperature, topographic conditions, water resources, and vegetation. The second part describes the analysis of the socioeconomic conditions for ski area development. Socioeconomic suitability was evaluated based on the linear weighting of four socioeconomic indexes: economic conditions, distance to a city, accessibility of transportation, and distance to a tourist attraction. Subsequently, the spatial distribution of the natural suitability and socioeconomic suitability indexes were used to obtain the integrated suitability index, which identifies the areas with the greatest potential for ski area development. Finally, we analyzed the fundamental driving factors of ski area development based on natural and socioeconomic conditions.

### 3.1 Natural suitability

For ski area development, natural suitability shapes ski areas from the supply side (Fig. 3). In general, areas with high scores were characterized by an excellent natural endowment, including abundant snow resources, favorable temperature, suitable terrain, beautiful natural landscape, and adequate water resources. The areas with high scores were widely distributed, mainly including the Changbai Mountains and Xiaoxing'an Mountains (northeast China), the Daxing'an Mountains (Inner Mongolia Province), the Tianshan Mountains and the Altai Mountains (Xinjiang Province), the Qilian Mountains (Gansu Province and Qinghai Province), the Yanshan Mountains (Beijing-Tianjin-Hebei region), the Lvliang Mountains and the Taihang Mountains (Shanxi Province), the southeastern Tibetan Plateau (Sichuan Province), and the Qinling Mountains (Shaanxi Province).

Areas with high scores were always located in mountains, particularly in northwestern and northern China. Additionally, as mentioned in Sect. 2, most areas of the Tibetan Plateau were unsuitable for ski area development because the elevation was too high. There was also a large area covering the Gobi Desert in the middle region of northwestern China with low natural suitability values. With the increase in temperature from north to south, ski areas need to be built in mountains with high elevations. High temperatures led to the low natural suitability values in southern China. In the eastern part of China, the low values were due to poor topographic conditions (flat terrain).

### 3.2 Socioeconomic suitability

For ski area development, socioeconomic suitability plays a crucial role by evaluating ski areas from the demand side (Fig. 4). The regions given high scores in this figure had favorable socioeconomic conditions. For instance, these areas were usually close to well-developed regions, markets, roads and tourist attractions with better economics and infrastructure. Ski areas near regions with high socioeconomic suitability values would give enterprises a more advantageous environment.

Figure 4 shows the significantly imbalanced socioeconomic development between eastern and western China. The socioeconomic suitability of western China lags far behind that of eastern China. This pattern differs from the spatial

distribution of natural suitability. The regions with high socioeconomic suitability values were mainly distributed in the eastern coastal areas, including the Yangtze River Delta, the belt encircling the Bohai Sea (including the Beijing-Tianjin-Hebei region), followed by the urban agglomerations of the Central Plains Economic Zone and the Northeast Economic Zone. Additionally, the Hohhot-Baotou-Yinchuan-Yulin Economic Zone is an important hub connecting northern and western China, and the Northern Tianshan Mountain Economic Zone is the largest economic belt in western China, which is located in the core area of the Belt and Road.

### 3.3 Integrated suitability

For ski area development, integrated suitability was evaluated based on the spatial distribution of natural and socioeconomic suitability (Fig. 5). The integrated suitability results identified ten regions that have the greatest potential for ski area development, which were the Changbai Mountains (northeast China), the Daxing'an Mountains (Inner Mongolia Province),the Yanshan Mountains (Beijing-Tianjin-Hebei region), the Lvliang Mountains and the Taihang Mountains (Shanxi Province), the Tianshan Mountains (the Northern Tianshan Mountain Economic Zone), the Qilian Mountains (Qinghai Province and Gansu Province), the Qinling Mountains (Shaanxi Province), the area surrounding Mount Tai (Shandong Province), the Southeast hills (Yangtze River Delta), and the southeastern Tibetan Plateau (Sichuan Province).

We further classified each of the natural suitability (Fig. 3) and socioeconomic suitability (Fig. 4) domains into deciles and then cross-tabulated them to generate 100 unique combinations marked with gradient colors (Fig. 6). In this scheme, the red-shaded areas are where socioeconomic suitability is high but natural suitability is modest. In contrast, the green-shaded areas have high natural suitability values but low socioeconomic suitability values. The black-shaded areas have high values for both types of suitability, whereas the yellow-shaded areas have low values for both types of suitability.

Figure 6 illustrates the fundamental driving factors of ski area development in China. The analysis identified four zones. The first zone consists of "natural-driven areas" with favorable natural conditions but relatively poor socioeconomic status. The second zone consists of "socioeconomic-driven areas", which are characterized by a good socioeconomic status but relatively modest natural conditions. The third zone consists of "ideal areas", which feature both favorable natural conditions and advantageous socioeconomic conditions. Finally, the fourth zone consists of "unfavorable areas", which have poor natural and socioeconomic conditions. In total, 35.4 million hectares (9.2% of the analyzed area) were categorized as natural-driven areas (Table 4), which were mainly distributed in the mountains of northwestern and northeastern China and the area around the Tibet Plateau. A total of 183.7 million hectares (47.7% of the analyzed area) were classified as socioeconomic-driven areas, including large areas of eastern and central China. A total of 46.1 million hectares (12% of the analyzed area) were determined as ideal areas, particularly distributed in the Beijing-Tianjin-Hebei region, the eastern part of northeast China and central Xinjiang Province. Finally, 120.2 million hectares of land (31.2% of the analyzed area) were found to be unfavorable areas, which were mainly distributed in regions of northwestern and southern China. The excluded areas (because of high temperature and high elevation) account for more than half of the total area of China.

Due to the influence from socioeconomic conditions, the integrated suitability was weakened in the areas of the Altai Mountains, the Daxing'an Mountains and the marginal zone of the Tibetan Plateau. In contrast, the integrated suitability of eastern China has benefited from better socioeconomic conditions, as in Shandong Province, the Yangtze River Delta and the Beijing-Tianjin-Hebei region (Fig. 5).

## 4 Discussion

This section first used field survey information to validate the evaluation method, and the method was used to evaluate the locational suitability of ski areas established after 2012. Second, according to the validation and evaluation results, a series of development strategies were proposed. Finally, the limitations of this work were discussed.

### 4.1 Validation and evaluation

The existing 598 ski areas were illustrated on the map of the spatial distribution of the integrated suitability for ski area development (Fig. 7), including the ski areas used as samples (116), the ski areas established after 2012 (447) and surveyed (35). In addition, to evaluate the locational suitability of existing ski areas, Fig. 8 presents a detailed analysis of the natural suitability, socioeconomic suitability and integrated suitability of ski areas established after 2012.

To validate the suitability evaluation method, we analyzed the 35 ski areas that were investigated in the field (Fig. 7). The results show that the 5S-graded (highest grade) ski areas are clustered in areas with high integrated suitability values (ideal area), which are mainly distributed in high latitude regions, such as northeast China, the Beijing-Tianjin-Hebei region and Xinjiang Province; the 4S-graded and 3S-graded (medium grades) ski areas are distributed around Beijing and the marginal zone of the Tibetan Plateau; and the 2S-graded and 1S-graded (lowest grades) ski areas are mainly located in southern China, the North China Plain and western Xinjiang Province (Fig. 7). In general, the higher the grade of the ski area is, the better its locational suitability. In most natural-driven areas, such as the western Xinjiang Province, small ski areas are established rather than large resorts due to the low socioeconomic suitability. There are a number of low-grade ski areas with poor facilities in low elevation and midlatitude regions, such as the North China Plain and the Southeast hills (socioeconomic-driven areas). In the North China Plain, according to the interviews with ski area managers, vicious competition between enterprises due to similar products becomes more serious that it is difficult for small ski areas to continue operations. Additionally, in southern China, selected ski areas are operated by the government to promote winter sports and even run without profit.

As shown in Fig. 8, 25.99% of the ski areas fall within ideal areas, 69.84% of the ski areas are situated in socioeconomic-driven areas, 1.86% of the ski areas are located in natural-driven areas, and 2.32% of the ski areas are distributed in unfavorable areas. Additionally, the areas are categorized as having low, medium and high suitability according to the integrated suitability value. After 2012, 89.1% of the ski areas were distributed in medium suitability areas, whereas 10.44% of the ski areas were located in high suitability areas. There were 2 ski areas in low suitability areas. In total, 84.23% of the ski areas were located in areas with an integrated suitability value greater than 0.5, while 15.77% of the ski areas were located in areas with integrated

suitability values less than 0.5, which are almost all distributed in socioeconomic-driven areas. Notably, the ski areas in socioeconomic-driven areas are more prone to environmental problems due to the decreased natural suitability, and these enterprises may soon face the challenges of dismal prospects.

A field survey of the ski areas revealed that the suitability results adequately reflect the actual conditions for ski area development. Additionally, the evaluation results indicate that some small ski areas have been built in areas with low integrated suitability values, with most being distributed in socioeconomic-driven areas.

## 4.2 Development strategy

To meet the goal of having 300 million Chinese citizens involved in winter sports by 2022, China is making an enormous effort to popularize winter sports by improving the accessibility of facilities and training for beginners. According to China's

General Sports Administration's "13th Five-Year Plan", which is a roadmap for development from 2016 to 2020, snow and ice sports should be vigorously promoted, especially in southern and northwestern China.

In fact, northwestern China has abundant snow and ice resources. Thus, in this less developed area, the vigorous development of ski tourism is an opportunity to promote regional economic development. In contrast, the favorable economic conditions in southern China can promote ski tourism development. The expansion of snow and ice sports in southern China will make

winter sports more accessible for citizens living in warm climates, which will bring a large number of new amateurs to the sport.

However, the poor economic conditions in northwest China and the low natural suitability in southern China, which can restrict the development of ski areas, should be considered. Especially in southern China, as noted in Sect. 4.1, there are a few small ski areas due to low natural suitability, and enterprises face similar products and fierce competition. Under global warming,

increased winter temperatures and decreased snowfall will severely affect these ski areas (Fukushima et al., 2002; Dawson, Scott, and McBoyle, 2009), causing economic losses, resource waste and environmental damage.

To stimulate the sustainable development of the ski industry, the location of ski areas should carefully be determined according to the results of the suitability evaluation. Based on our analysis, corresponding development strategies for the development of the Chinese ski tourism industry are proposed:

(1) In natural-driven areas, upscaled ski areas mainly used for competition and training should be built, which can improve the popularity of ski areas by hosting sporting events. Several small and medium-sized ski areas with good snow quality should also be appropriately planned around cities.

(2) In socioeconomic-driven areas, the ski area markets are determined by local visitors. According to the situation of local socioeconomic development, an appropriate number of small ski areas are advised to be built for recreational sports to expand

the influence of snow sports. However, a large number of small-size ski areas with poor facilities have been established in the economically driven areas, and these ski areas are unsafe and provide skiers with low-quality ski experiences. Therefore, in socioeconomic-driven areas, the number of ski areas should be limited, and enterprises should enhance their competitiveness by improving the quality of the ski area.

(3) In ideal areas, multiple high-end ski resorts with complete support facilities can feasibly to attract national and international visitors by holding major international sporting events.

(4) Unfavorable areas with both poor natural and socioeconomic conditions are unsuitable for skiing.

(5) In addition, in economically developed southern China, snow resources are insufficient because of high temperatures; thus, to popularize snow recreation, indoor ski slopes can be considered. Furthermore, in the less developed and oxygen-deficient Tibet Plateau, commercial ski areas are difficult to operate.

## 4.3 Limitation of the current work

In the current study, for the index of snow cover, only natural snow has been considered, while machine-made snow was represented in the form of air temperature (as mentioned in Sect. 2.1.1), which is indeed a proxy for potential snowmaking capacity. If the air temperature-based proxy model for snowmaking was used instead of the physics-informed model considering the snowpack, the length of the ski season may be underestimated (Scott et al., 2019), which is considered as a limitation of the current research. In addition, since the exact number of skiable days cannot be obtained from the proxy model, the viability of a specific ski area cannot be exactly estimated (Steiger et al., 2017). Previous studies employed forcing data from ski areas or nearby meteorological stations to calculate the length of the ski season for analyzing climate change vulnerability for specific ski destinations (Scott, McBoyle, and Mills, 2003; Hennessy et al., 2008; Steiger, 2010; Pons et al., 2015). However, the main barrier that arises when applying such kind of site-scale snow models to a large scale (e.g., the national scale in our study) is associated with the difficulties in obtaining high-quality data with fine spatial and temporal resolution and the heterogeneous spatio-temporal distribution of snow across scales. In particular, we would like to discuss the applicability of using the well-acknowledged ski season simulation model (e.g., SkiSim 2) in our research as an example. SkiSim 2 combines both the natural and machine-made snowpack models, and is widely applied in vulnerability assessment for ski areas (Steiger et al., 2017). There are two key parameters (snowfall temperature and degree-day factor) in SkiSim 2 need to be calibrated specifically for each ski area (Steiger, 2010; Scott et al., 2019), which can be obtained from at-site weather stations. It is noted that most ski areas in China are still under-development and not necessarily equipped with meteorological observations or close to weather stations, which hampers researchers from obtaining qualified meteorological data for parameterization in site-scale models (such as SkiSim 2). Another example is that Hennessy et al. (2008) used hourly wet-bulb temperature to model the snowmaking potential for Australian ski areas, and they computed the volume of machine-made snow based on the difference between the target snow depth and natural snow depth cover a typical ski run. Again, unfortunately, hourly wet-bulb temperature data are not available for most ski areas in China, let alone the grid data over entire region of China. In summary, previous studies using snowmaking models have been well-acknowledged at the site-scale. However, since the aim of this study was to provide the guidelines for the ski market at the national scale rather than debating whether a specific resort would be viable, we believe that using the air temperature-based proxy models to reflect the snowmaking conditions is acceptable over large-scale regions.

The second limitation stems from the air temperature data. Subzero wet-bulb temperatures are needed for snowmaking (Hennessy et al., 2008). Unfortunately, meteorological stations do not provide wet-bulb temperature data. As a proxy, the dry-bulb temperature is also widely used in machine-made snow models (Scott, McBoyle, and Mills, 2003; Steiger, 2010). Therefore, in this study, the dry-bulb temperature data were used to generate the index of air temperature, which reflects the snowmaking conditions and tourist comfort.

Another limitation is associated with wind, which has a significant impact on the efficiency of snowmaking and the attractivity of a site (Spandre et al., 2017). Wind speeds higher than 6 m s$^{-1}$ have a direct physical or mechanical effect on tourists (De Freitas, 2005). Daily wind speed data are available from the meteorological stations that are generally located around a city, while most ski areas are built in the mountains far from cities. Hence, the relatively sparse meteorological stations across the study area did not allow for the creation of reliable daily gridded data. In addition, wind is considered one of the most difficult meteorological variables to model due to its dependence on the specific characteristics of any given location, such as topography and surface roughness (Morales, Lang, and Mattar, 2012). The local micrometeorology at a specific location also cannot be captured using existing reanalysis wind speed data with low resolution.

The method of this work attempts to identify the suitability patterns of the main ski areas in China, but a more detailed and refined analysis based on local data will be necessary before deciding to invest (or not) in a new ski area. Based on previous studies, machine-made snow, air temperature and wind should be addressed in future studies on specific ski areas.

**5 Conclusion**

This study integrated natural and socioeconomic conditions from the supply and demand perspectives to quantitatively evaluate the locational suitability for ski area development in China. Five corresponding evaluation indexes of natural conditions were proposed: snow cover, air temperature, topographic conditions, water resources, and vegetation. The socioeconomic conditions included four indexes: economic conditions, accessibility of transportation, distance to a tourist attraction and distance to a city. Using GIS spatial analysis technology combined with remote sensing, online and field survey data, we presented a linear weighted method for synthetic evaluation in which the weight coefficients were calculated using an objective method based on entropy weight theory. The results showed the spatial distribution of the locational suitability of ski areas and identified the areas with the greatest potential for ski area development. Four zones were also identified by analyzing driving factors based on natural and socioeconomic conditions. Additionally, our method to estimate the suitability was validated based on field surveys of ski areas and was used to evaluate the locational suitability of ski areas established after 2012. Finally, corresponding development strategies for decision-makers were proposed.

Despite that there were studies using physics-informed snowmaking models to assess the vulnerability for specific ski areas, the aim of this study was to provide the scientific metrics for the ski market at the national scale. Even though the current model used a proxy for snowmaking with certain limitations (such as not counting the exact number of skiable days), this

study is feasible at the current scale of interest and expected to pave the way for further site-scale research (e.g., specific ski areas) with more detailed and refined analyses based on local data and other sources of information.

The results of the weight coefficients indicate that snow resources, air temperature and topographic conditions are major natural factors that influence ski area development. In the context of global warming, it is necessary to evaluate the vulnerability of a ski area to future climate change (Steiger et al., 2017). With the increase in winter temperatures and the decrease in snowfall in the next few decades (Ji and Kang, 2012; Zhou, et al., 2018), the natural suitability values for southern and eastern China will decrease, and small midlatitude and low-elevation ski areas will be the first to close due to poor snow conditions (Bark, Colby, and Dominguez, 2010; Gilaberte-Búrdalo et al., 2017). Additionally, with social and economic development, people's living standards will greatly improve. Thus, the socioeconomic suitability in northwestern and northeastern China may increase. As a result, northwestern and northeastern China may become popular markets and central places for ski tourism. To more thoroughly study the future of ski tourism in China, future research is needed to evaluate the locational suitability of ski areas in relation to climate change and socioeconomic development.

*Data availability.* The MODIS snow cover products are available for downloaded from the National Snow Ice Data Center (NSIDC, https://nsidc.org/). The SD products were downloaded from the Cold and Arid Regions Sciences Data Center at Lanzhou (http://westdc.westgis.ac.cn). The air temperature dataset was developed by Wu and Gao (2013). The VIIRS DNB nighttime data are available from https://www.ngdc.noaa.gov/eog/viirs/download_dnb_composites.html.

*Author contributions.* TC and JD conceived this study with input from CX, LD and AM. SW led the field surveys of ski areas. The manuscript was written by JD and TC. All authors contributed to editing and revision.

*Competing interests.* The authors declare no conflict of interest.

*Acknowledgments.* We thank X. F. Yang for fruitful discussions and two reviewers, C. M. Carmagnola and D. Scott, for their thorough reviews and insightful comments to improve the manuscript.

*Financial support.* This study was supported by the Chinese National Natural Science Foundation (41690140 & 41671058) and the National Foundational Scientific and Technological Work Programs of the Ministry of Science and Technology of China (grant nos. 2017FY100501).

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

List of Figure captions

Figure 1. The spatial distribution of the natural condition indexes and excluded areas.

Figure 2. The spatial distribution of the socioeconomic condition indexes.

Figure 3. The spatial distribution of natural suitability for ski area development based on snow cover, air temperature, topographic conditions, water resources, and vegetation.

Figure 4. The spatial distribution of socioeconomic suitability for ski area development based on economic conditions, distance to a city, accessibility of transportation, and distance to a tourist attraction.

Figure 5. The spatial distribution of integrated suitability for ski area development based on the natural and socioeconomic indexes.

Figure 6. Spatial distribution of the driving factors of ski area development.

Figure 7. The distribution of existing ski areas in China.

Figure 8. The suitability values of existing ski areas established after 2012.

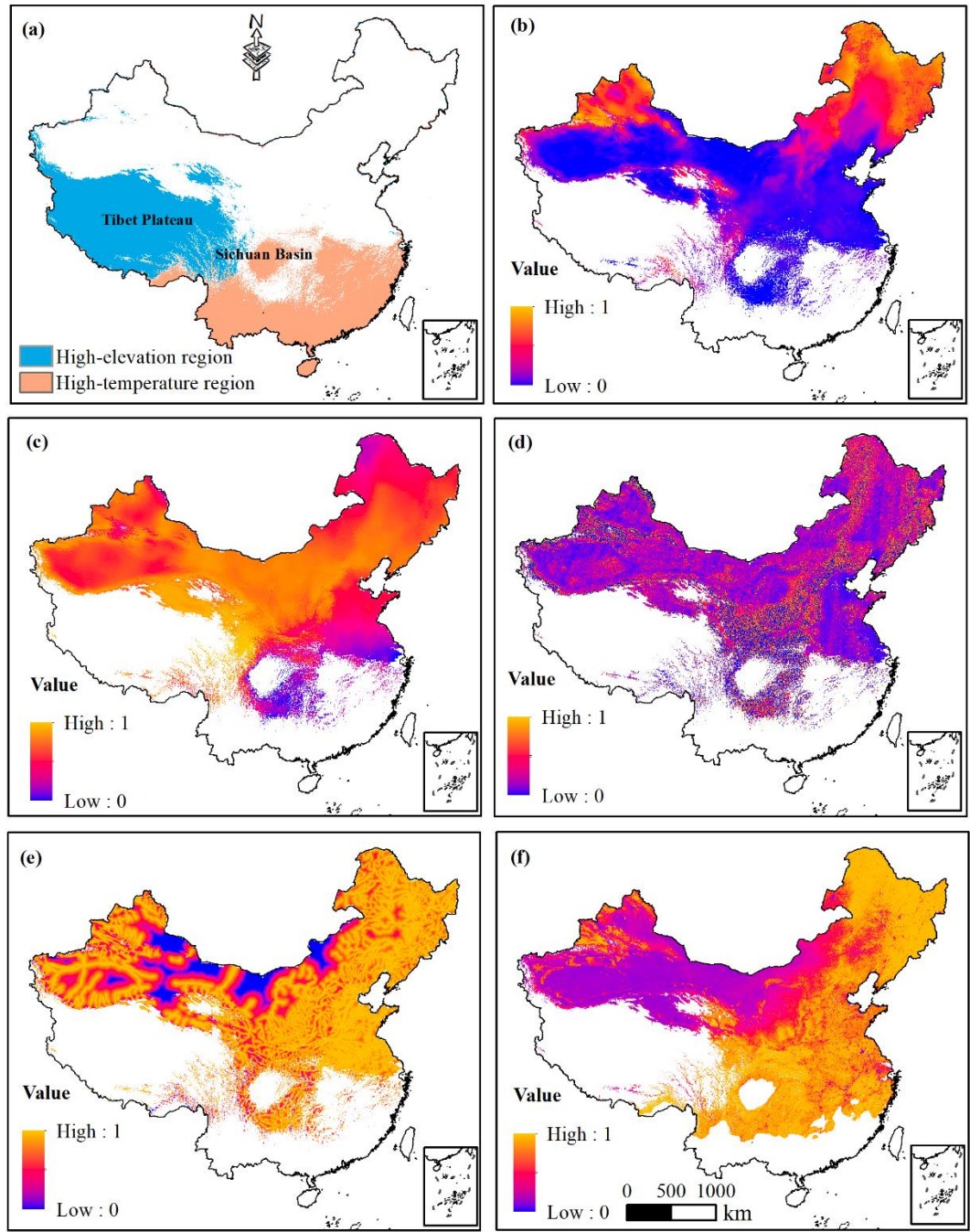

**Figure 1. The spatial distribution of the natural condition indexes and excluded areas. (a) Excluded areas, (b) snow cover, (c) air temperature, (d) topographic conditions, (e) water resources, (f) vegetation.**

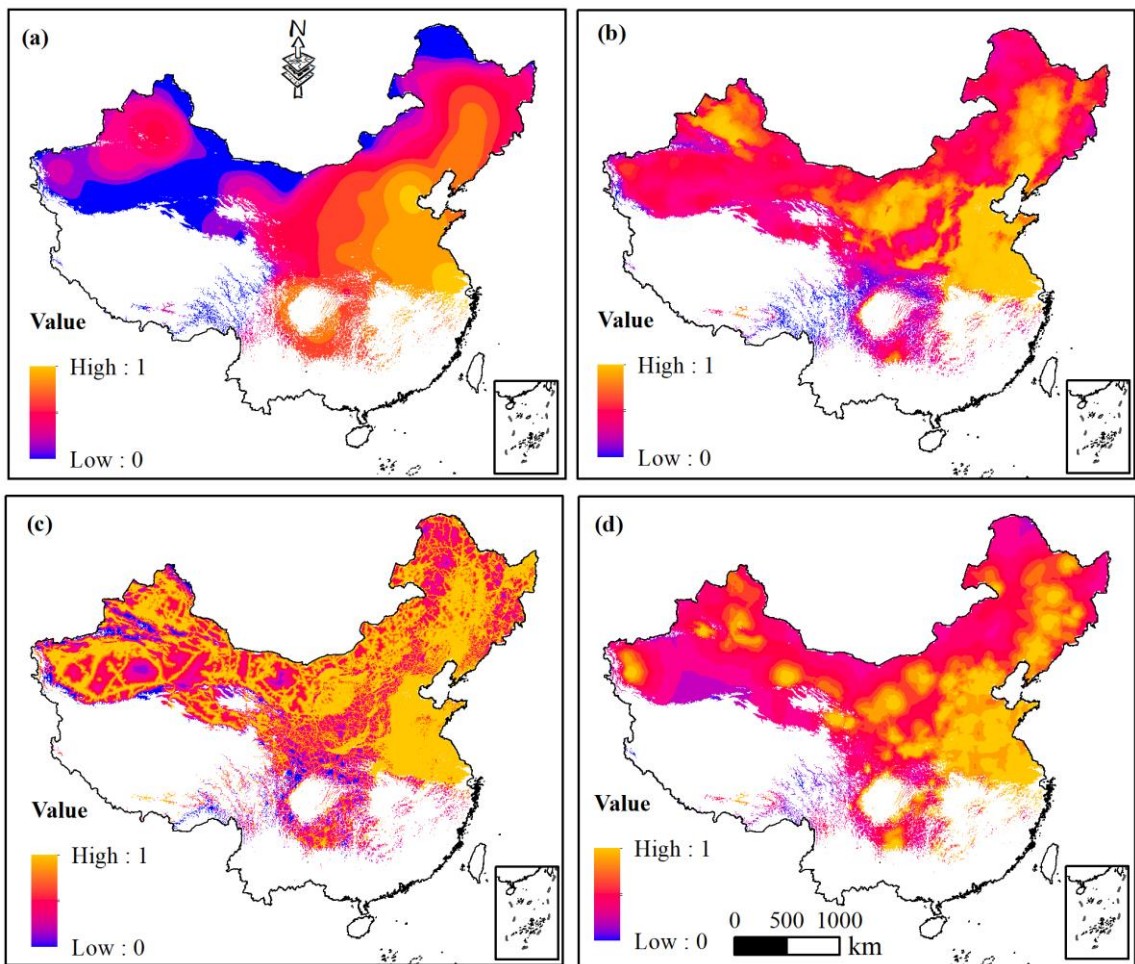

**Figure 2. The spatial distribution of the socioeconomic condition indexes. (a) Economic conditions, (b) distance to a city, (c) accessibility of transportation, (d) distance to a tourist attraction.**

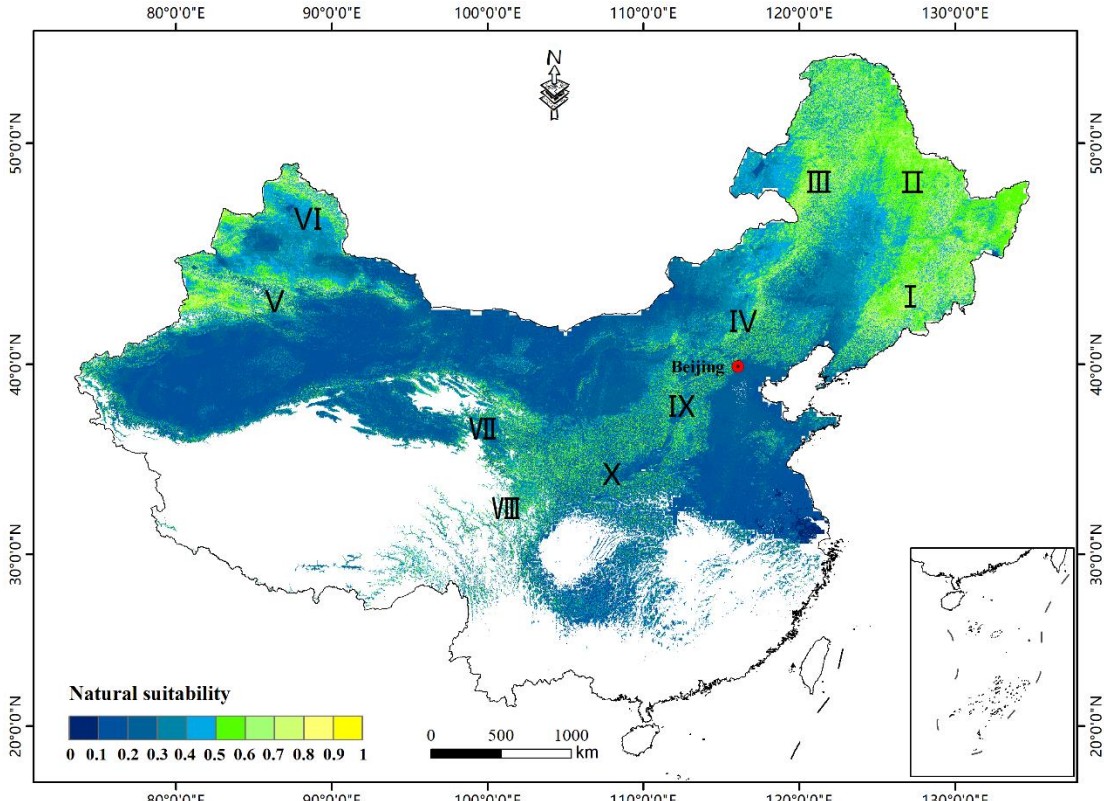

**Figure 3. The spatial distribution of natural suitability for ski area development based on snow cover, air temperature, topographic conditions, water resources, and vegetation. I: Changbai Mountains; II: Xiaoxing'an Mountains; III: Daxing'an Mountains; IV: Yanshan Mountains; V: Tianshan Mountains; VI: Altai Mountains; VII: Qilian Mountains; VIII: southeastern Tibetan Plateau; IX: Lvliang Mountains and Changbaishan Mountains; X: Qinling Mountains.**

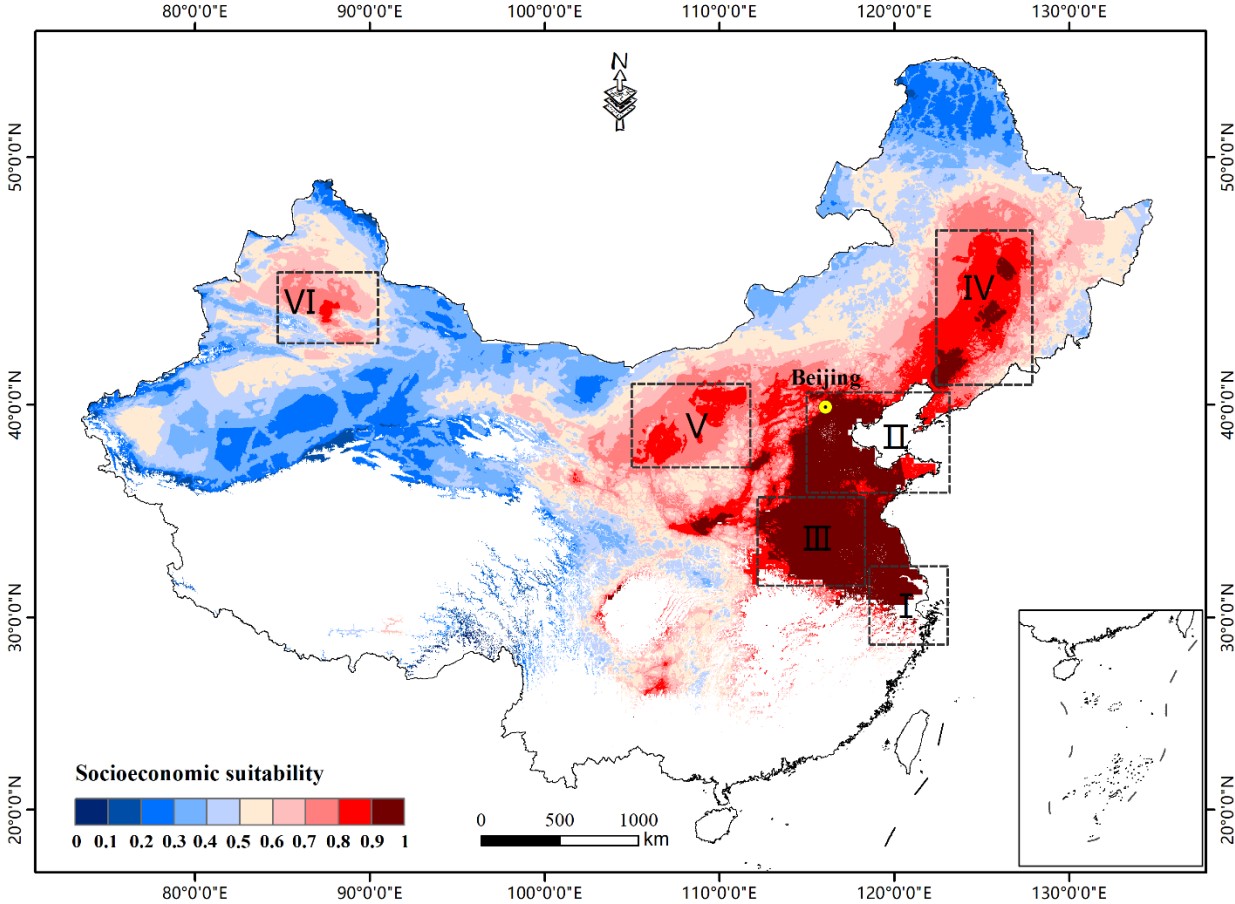

**Figure 4. The spatial distribution of socioeconomic suitability for ski area development based on economic conditions, distance to a city, accessibility of transportation, and distance to a tourist attraction. I: Yangtze River Delta; II: belt encircling the Bohai Sea (including the Beijing-Tianjin-Hebei region); III: Central Plains Economic Zone; IV: Northeast Economic Zone; V: Hohhot-Baotou-Yinchuan-Yulin Economic Zone; VI: Northern Tianshan Mountain Economic Zone.**

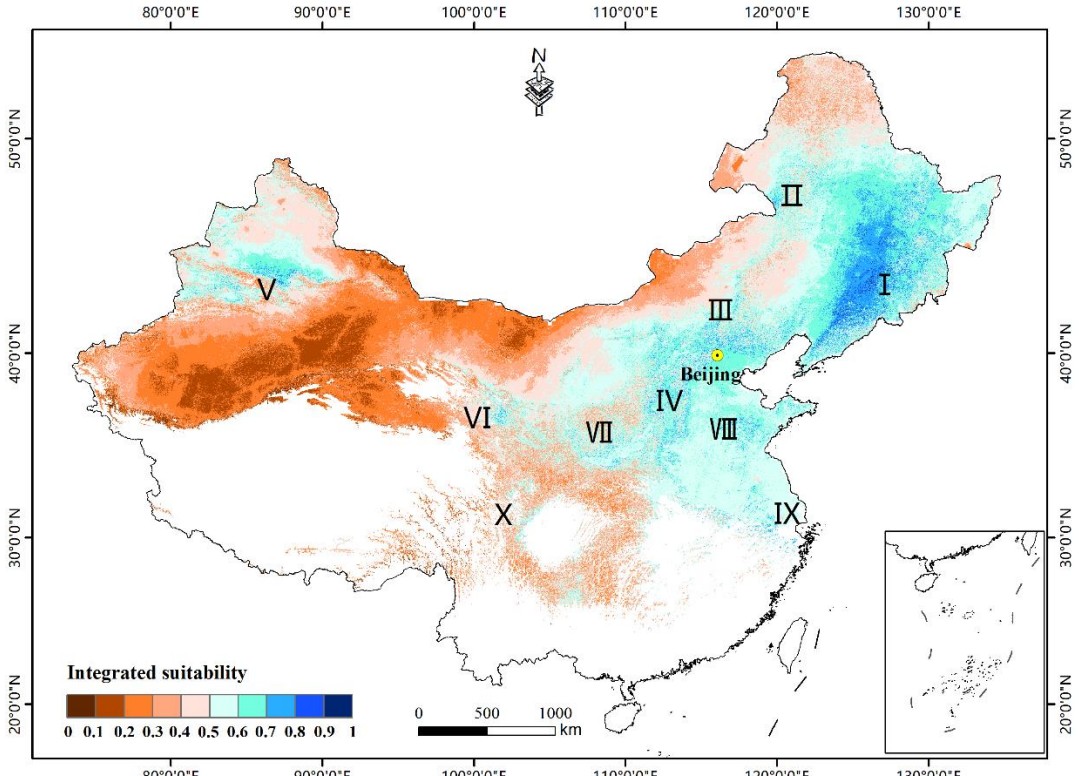

**Figure 5. The spatial distribution of integrated suitability for ski area development based on the natural and socioeconomic indexes.**
**I: Changbai Mountains (northeast China); II: Daxing'an Mountains (Inner Mongolia); III: Yanshan Mountains (Beijing-Tianjin-Hebei region); IV: Lvliang Mountains and Taihang Mountains (Shanxi Province); V: Tianshan Mountains (the Northern Tianshan Mountain Economic Zone); VI: Qilian Mountains (Qinghai Province and Gansu Province); VII: Qinling Mountains (Shaanxi Province); VIII: the area surrounding Mount Tai (Shandong Province); IX: Southeast hills (Yangtze River Delta), X: southeastern Tibetan Plateau (Sichuan Province).**

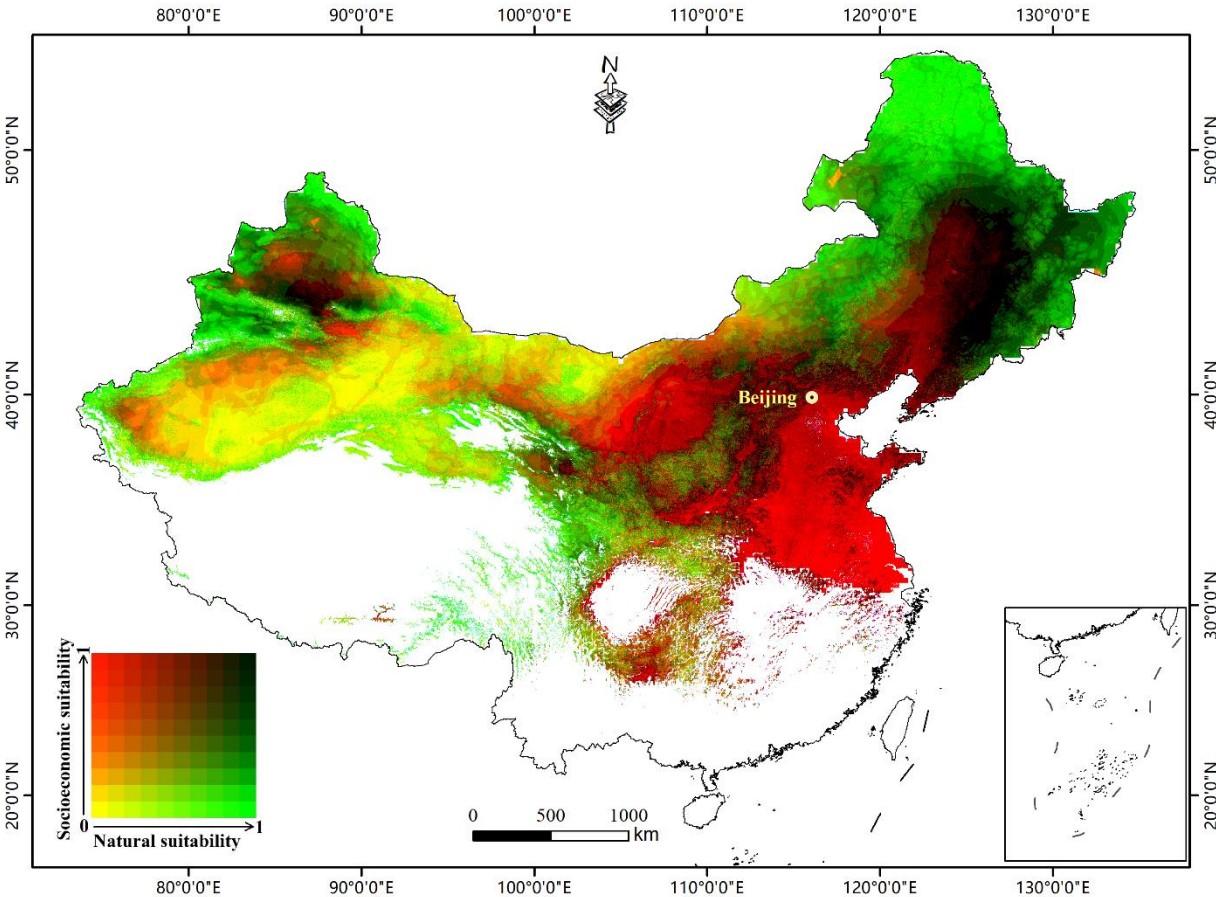

**Figure 6. Spatial distribution of the driving factors of ski area development. The natural suitability and socioeconomic suitability values are classified into deciles, generating 100 unique color combinations. Socioeconomic-driven areas are shown in red shades, natural-driven areas in green shades, ideal areas in black shades, and unfavorable areas in yellow shades.**

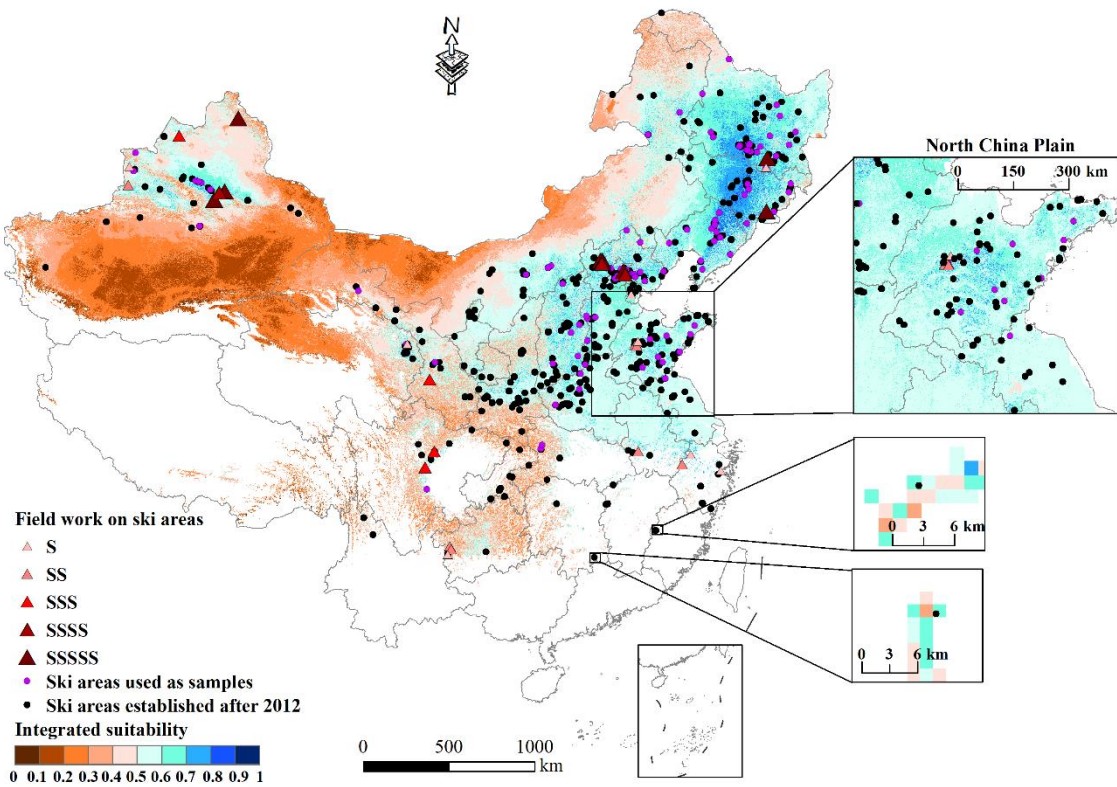

**Figure 7. The distribution of existing ski areas in China. The ski areas used as samples are marked with purple dots. The ski areas established after 2012 are marked with black dots. The ski areas that were investigated in the field are marked with triangles, and the larger the triangle is, the higher the grade of the ski area. Details on the North China Plain and southern China are shown in the black insets (right).**

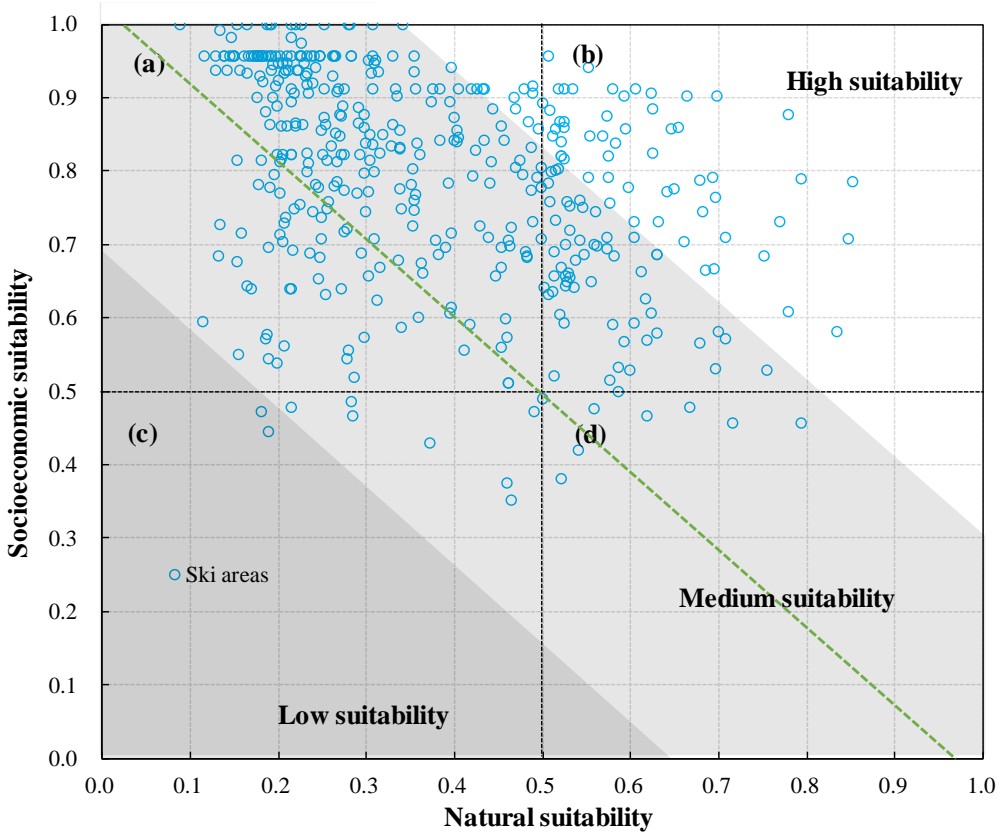

**Figure 8. The suitability values of existing ski areas established after 2012. (a) Socioeconomic-driven areas, (b) ideal areas, (c) unfavorable areas, (d) natural-driven areas. The deeply shaded region has integrated suitability lower than 1/3 (low suitability), the lightly shaded region has integrated suitability greater than 1/3 and lower than 2/3 (medium suitability), and the white region has integrated suitability greater than 2/3 (high suitability). The green dashed line is the demarcation line with an integrated suitability of 0.5. The shaded regions and the green line do not intercept the x and y axes at the same point because natural suitability and socioeconomic suitability have different weight coefficients (0.52 and 0.48, respectively).**

List of Table captions

Table 1. Descriptions of the data used in this study.

Table 2. Different daily air temperature regimes and corresponding scores.

Table 3. Weight coefficients for the evaluation indexes.

5    Table 4. Surface areas and driving factors for four zones of suitability.

**Table 1. Descriptions of the data used in this study.**

| Data | Period | Spatial attribution | Source |
|---|---|---|---|
| Moderate-resolution imaging spectroradiometer (MODIS) snow cover products | 2010-2015 | 500 m | National Snow Ice Data Center, https://nsidc.org/ |
| Snow depth (SD) products | 2010-2015 | 25 km | Cold and Arid Regions Sciences Data Center, http://westdc.westgis.ac.cn |
| Gridded daily observation dataset | 2010-2015 | 25 km | Wu and Gao (2013) |
| Shuttle Radar Topography Mission (SRTM) digital elevation model (DEM) (V004) | 2000 | 90 m | National Map Seamless Data Distribution Systems, http://seamless.usgs.gov |
| Annual normalized difference vegetation index (NDVI) spatial distribution dataset | 2015 | 1 km | Data Center for Resources and Environmental Sciences, http://www.resdc.cn |
| Annual average radiance composite images | 2015 | 500 m | National Oceanic and Atmospheric Administration's National Geophysical Data Center, https://www.ngdc.noaa.gov/eog/viirs/download_dnb_composites.html |
| River spatial distribution data | 1999 | line | Data Center for Resources and Environmental Sciences, http://www.resdc.cn |
| Lake spatial distribution data | 2010 | polygon | Data Center for Resources and Environmental Sciences, http://www.resdc.cn |
| Cities spatial distribution data | 2009 | point | Geographic Data Sharing Infrastructure, College of Urban and Environmental Science, Peking University, http://geodata.pku.edu.cn |
| Road network | 2015 | line | OpenStreetMap, https://www.openstreetmap.org/#map |
| List of public-use airports | 2018 | document | Civil Aviation Administration of China, http://www.caac.gov.cn |
| Tourist attractions | 2018 | point | Online map, https://maps.baidu.com |
| Existing (598) ski areas | 2018 | point | Online map, https://maps.baidu.com |
| 35 ski areas | 2018 | document | Field surveys |

**Table 2. Different daily air temperature regimes and corresponding scores.**

| Daily mean temperature (°C) | Score |
| --- | --- |
| -2 ~ -5 | 7 |
| -5 ~ -10 or -2 ~ 0 | 6 |
| -10 ~ -15 | 5 |
| -15 ~ -20 or 0 ~ 5 | 4 |
| -20 ~ -25 | 3 |
| -25 ~ -30 or 5 ~ 10 | 2 |
| < -30 | 1 |
| > 10 | 0 |

**Table 3. Weight coefficients for the evaluation indexes.**

| Indexes | Weight coefficients |
| --- | --- |
| Natural suitability | 0.52 |
| Snow cover | 0.36 |
| Air temperature | 0.19 |
| Topographic conditions | 0.32 |
| Water resources | 0.04 |
| Vegetation | 0.09 |
| Socioeconomic suitability | 0.48 |
| Economic conditions | 0.37 |
| Distance to a city | 0.29 |
| Accessibility of transportation | 0.18 |
| Distance to a tourist attraction | 0.16 |

**Table 4. Surface areas and driving factors for four zones of suitability.**

| Zones | Natural suitability | Socioeconomic suitability | Area (ha×10$^6$) | Percentage (%) |
|---|---|---|---|---|
| Natural-driven areas | 0.5~1 | 0~0.5 | 35.4 | 9.2 |
| Socioeconomic-driven areas | 0~0.5 | 0.5~1 | 183.7 | 47.7 |
| Ideal areas | 0.5~1 | 0.5~1 | 46.1 | 12 |
| Unfavorable areas | 0~0.5 | 0~0.5 | 120.2 | 31.2 |