# Peer review of "Suitability Analysis of Ski Areas in China: An Integrated Study Based on Natural and Socioeconomic Conditions"

_The Cryosphere, 2019_

## Referee Comment (RC1) · Carlo Maria Carmagnola (Referee) · 2 Apr 2019

**Review**

Jie Deng et al., *Suitability Analysis of Ski Areas in China: An Integrated Study Based on Natural and Socioeconomic Conditions*, The Cryosphere Discuss., https://doi.org/10.5194/tc-2019-43

Reviewer: C. M. Carmagnola

This study presents a method to evaluate the suitability of existing and future ski areas in China based on a combination of natural and socioeconomic conditions. Blending geographic information system spatial analysis, remotely-sensed observations, online and field data, the authors were able to define an integrated index to identify the best suited locations for ski-resorts in the country. This index was evaluated using the locations of already existing resorts and more detailed information from a field survey of a panel of 35 resorts. Results show that 92% of the existing ski-resorts are currently located in areas with medium-to-high suitability indexes. As for future ski-resorts, this analysis framework could help driving new investments in the ski market at the national scale, by defining objective indicators to identify regions potentially interesting for developing ski-related activities. To this aim, the authors also suggest some broad strategies to guide the next steps of ski tourism industry in China.

The main novelty of this work lies in the method used for aggregating the different natural (snow cover, air temperature, topography, groundwater, vegetation) and socioeconomic (welfare, accessibility of transportation, distance to tourist attractions, distance to cities) conditions. Applying a linear weighting method allowed the authors to compute the relative contributions of each condition to the integrated suitability index. However, this part of the work also represents the weakest section of the paper in its current form. Indeed, the approach followed to compute and then aggregate the weight coefficients starting from a data sample of 128 ski-resorts is not presented clearly. In particular, it appears that the same dataset has been used for informing the aggregation method and to evaluate its results. Moreover, and perhaps more importantly, the computation of individual indexes, namely those for snow cover and air temperature, raises some concerns. In this regard, for example, is it not clear how the snow depth and snow cover days have been estimated and there seems to be some confusion between air temperature and wet bulb temperature, which is the main parameter for snowmaking. More generally, I would not overstate the importance of the main results of this study for a direct application. Even if the method can be seen as a first attempt to identify the main ski area's suitability pattens in China, a more detailed and refined analysis based on local data will be necessary before deciding to invest (or not) in a new ski-resort. In other words, the results of this study provide interesting guidelines for the ski market at the national scale,

but do not tell whether a specific resort will be, in fact, viable.

Overall the paper is easy to follow and well written, beside some small mistakes and sentences that need to be rephrased. In light of the novel approach and the great effort put by the authors in combining and homogenizing data coming from various sources in a consistent and coherent way, I support publication in The Cryosphere. However, several points should be considered to improve the presentation and some major issues should be addressed before publication can be recommended.

P.1 - L.19: "To evaluate" → "Therefore, evaluating".

P.1 - L.20: "has since become" → "has become".

P.1 - L.21: "using linear" → "using a linear".

P.1 - L.22: "information systems" → "information system".

P.1 - L.25-28: I would reverse the order of two sentences and rephrase them as follows: "As such, a metrics ranging from 0 to 1 considering both natural and socioeconomic conditions is used to define a suitability threshold for each candidate region for ski area development. A ski area is considered to be a dismal prospect when the locational integrated index is less than 0.5. The results show that 92% of existing ski areas are located in areas with an integrated index greater than 0.5".

P.2 - L.19: "freshwater resource" → "resource".

P.2 - L.22: On the effect of climate change on future snow conditions, you could also make reference to this recent work: Verfaillie et al. (2018).

P.2 - L.25: "particularly in lower elevation and mid-latitude ski areas" → "particularly at lower elevations and in mid-latitude ski areas".

P.2 - L.25-26: I do not entirely agree with this statement. In some cases, even at medium-to-long term, snowmwaking could prove to be useful for keeping ski areas reliable, compensating for the scarcity of natural snow (Spandre et al., 2018). While stressing very well the environmental impact related to snowmaking installations, this section does not focus enough on the added-value of machine-made snow production as a necessary complement to natural snow for ski-resorts. In this regard, you could see for example Steiger (2010), Hanzer et al. (2014) and Damm et al. (2014, 2017).

P.2 - L.30-31: These figures are a bit low. One ha of slope, covered with 30 cm of artificial snow having a density of 500 kg m$^{-3}$, will necessitate a water consumption of 1.5 million litres. However, snowmaking efficiency in converting water into snow is less than 80%. This is mainly determined by wind conditions, according to, for instance, Spandre et al. (2017). Therefore, in practice at least 2 million liters of water are needed to cover 1 ha with a 30 cm thick snowpack. As for the electric consumption, it is usually accepted that the current snow-gun generation requires 2 to 3 kWh to produce 1 m$^3$ of snow.

P.2 - L.34: "became critical since inappropriate selections" → "is critical since inappropriate choices".

P.3 - L.1: "To evaluate" → "Evaluating".

P.3 - L.5: An important reference that could be used to provide more information on the state of the Chinese ski market is Vanat (2019), in particular for the number of skier visits. This recent book updates some figures of the previous "China ski industry white book 2017".

P.3 - L.6: "public to" → "public in China to".

P.3 - L.25: "Systems" → "System".

P.3 - L.26: "linear weighted" → "a linear weighted".

P.3 - L.29: In the introduction, it is worth mentioning the work of Demiroglu et al. (2019) and Demiroglu (2019), who have defined a Ski Climate Index (SCI) to estimate the overall suitability of Turkey for ski tourism. This index has been designed as a combination of several factors, among which snow reliability, land cover, aesthetics, market accessibility, comfort (sunshine, wind and temperature conditions) and even security. There are several similarities between your approach and Demiroglu's.

P.3 - L.30-33: I would rephrase this section to improve clarity: "This paper is organized as follows: in Sect. 2, we provide a description of the data and method. Sect. 3 presents the results. Section 4 evaluates the method, discuss its limitations and proposes suggestions for the development strategy. The final section contains a brief conclusion and discusses future work."

P.4 - L.13: Please consider adding a table listing the values of each natural and socioeconomic index, along with their normalization (this has been done only for air temperature, in Table 2). Having a complete table summarizing all the indexes will greatly improve the readability of this section.

P.4 - L.17: "both as supplement and attractions". This is not clear, please rephrase.

P.4 - L.23: "rule as the most" → "rule and is the most".

P.5 - L.3: This entire "Snow cover" section should be revisited, since several vital points are quite obscure. What is the period considered to define the winter season? What is the minimum snow depth used to define a snow cover day? What do you mean when saying "snow depth is only taken as a reference for the index of snow cover"? When applying the 100-day-rule, did you consider consecutive or non-consecutive days? What MODIS bands and what products did you use to retrieve the snow cover fraction? How were the SD and SCD indexes normalized? You should also highlight the fact that these

observations are for natural snow only and the possibility to add some machine-made snow is taken into account by the next index, that for air temperature.

P.5 - L.17: The clarity of this section needs to be improved. First, it is not clear how you have computed daily mean and maximum air temperatures starting from a dataset of daily observations. Second, you do not explain what period you have considered to rule out high-temperature regions and if you require the 90 days with temperature greater than 10°C to be consecutive. Third, you should clearly distinguish between air temperature and wet bulb temperature. The latter is what really matters for snowmaking, and in this case the colder is the temperature, the better is the snowmaking efficiency. For this reason, you should explain that the 11 temperature regimes and their corresponding scores are designed as a trade-off between colder temperatures needed to preserve the snowpack and to produce machine-made snow and warmer temperatures needed by skiers. Finally, the effect of wind, which is not at all considered in this study, should be at least briefly discussed, since it can have a significant impact on both the attractivity of a site and its potential to sustain snowmaking.

P.5 - L.21-24: Usually, the slope classification goes like this: very easy = 15% = ∼9°; easy = 20% = ∼11°; hard = 30% = ∼17°; very hard = 40% = ∼22°. Therefore, I would not exclude slopes between 10° and 20°. Please comment on that.

P.5 - L.27: What minimum and maximum values did you use to normalize the slope gradient?

P.5 - L.33: Why did you consider only rivers and not, for example, lakes?

P.6 - L.2: Could you give a few more details on the cost distance method you allude to?

P.6 - L.5: "Vegetation is a representative of an ideal environment and an important" → "Vegetation contributes to creating an ideal environment and is an important".

P.6 - L.7: About wind conditions, please see comment on P.5 - L.17.

P.6 - L.14: Please explain briefly how the "geometrical interval classification" has been applied in this case.

P.6 - L.28: "relative values". Relative to what?

P.6 - L.28: "without an estimation" → "without a direct estimation".

P.6 - L.30: Please explain briefly how the "kernel density analysis" has been applied in this case.

P.7 - L.2: Please explain briefly how the "distance decay theory" has been applied in this case.

P.7 - L.4: Did you apply any selection criterion regarding the size of the cities?

P.7 - L.12: Would it be possible to consider, in addition to the road network, also the distance to airports?

P.7 - L.21: How do you define a ski area? For instance, Vanat (2019), distinguishing between ski areas (designated place where one skis and that may not have lifts) and ski resorts (an organized ski area with more than four lifts), reports a total of 703 Chinese ski areas.

P.7 - L.25: It is not at all clear how the dataset coming from the 128 ski areas has been used to calculate the weight coefficients. See comment on Section 2.3.

P.7 - L.27: "five grades, of which 1S to 5S represent" → "five grades from 1S to 5S, representing".

P.8 - L.1-3: This sentence is not clear. Do you mean that only 27 areas out of 35 have been ranked, since the others are already listed in the Xinjiang Government Tourist Office? If it is the case, you could rephrase as follows: "According to the feedbacks from our field surveys, ski areas were roughly ranked into different grades, except for the 8 ski areas whose grades were already defined in the Xinjiang Government Tourist Office."

P.8 - L.4: Section 2.3 needs to be improved in several aspects.
- In Eq.1 and Eq.2, you introduce the weighting factors $W$, while in Eq.11 you compute the weight coefficients $w$. Are they not the same?
- In Eq.1 and Eq.2, $i$ and $j$ are used for the natural and socioeconomic conditions, respectively. Starting from Eq.4, however, the same indexes are used for the number of optional schemes and the number of evaluating indexes. This generates some confusion.
- In Eq.4, you should say what the optional schemes and the evaluating indexes are in your case. If $n$ is equal to 128 and $m$ is equal to 9, this has to be stated clearly.
- In Eq.5, the reason why you have to normalize is not presented clearly.
- In Eq.6 and Eq.7, what does $r$ stand for? Should it not be $x$ instead? And $x_{j,max}$ instead of $x_{max}$?
- In Eq.10, why should $P_{ij}$ be equal to 0?
More generally, the main missing point is the description of how the dataset coming from 128 ski areas has been used to compute $w$, $a$ and $b$. In particular, it seems that the same dataset has been used to compute the coefficients and then to evaluate them. Please clarify this point.

P.10 - L.2: "led" → "lead". Sometimes you use the past tense, other times you use the present simple. Please standardize this throughout the text.

P.10 - L.5: "that evaluates" → "by evaluating".

P.10 - L.10: "China, which differs" → "China. This patterns differs".

P.10 - L.16: You say that the suitability index allows to identify 7 regions, but then you list 8, and in Fig.5 they are 10. Please check this out.

P.10 - L.16: "have greatest" → "have the greatest".

P.10 - L.22-23: Reference to Fig.6 should be made at the end of the previous sentence (after "gradient colors").

P.10 - L.24: I would replace "dark-shaded" with "black-shades" and "light-shaded" with "yellow-shaded".

P.10 - L.31: These surface areas in ha seem to be out by a factor of 10. Please check them again.

P.11 - L.3: "Moreover" → "Finally".

P.11 - L.5: Adding up the 4 areas, you get less than half the total area of China. Do the excluded areas account for the rest? Please comment on that.

P.11 - L.7: The low suitability here appears to be mostly due to poor natural conditions.

P.11 - L.8-9: "In contrast, the integrated suitability of eastern China was enhanced; thus, Shandong province, the Yangtze River Delta and the Beijing-Tianjin-Hebei region were more pronounced (Fig. 5)." → "In contrast, the integrated suitability of eastern China has benefited from better socioeconomic conditions, as in the Shandong province, the Yangtze River Delta and the Beijing-Tianjin-Hebei region (Fig. 5)."

P.11 - L.11: "verify the evaluation method" → "evaluate".

P.11 - L.13: "Verification" → "Evaluation".

P.11 - L.17: "0.5. However" → "0.5, while".

P.11 - L.27: "selected" → "can consider".

P.11 - L.30: "high" → "medium-to-high".

P.12 - L.10: What do you mean by "product homogeneity"?

P.12 - L.15: "find" → "found". Verb tenses should be carefully checked to ensure consistency (see comment on P.10 - L.2).

P.12 - L.16: "result" → "results".

P.12 - L.22: According to Vanat (2019), there are currently about 12 million skiers in China. Is the goal of having 300 million skiers in 3 years realistic? Please comment.

P.12 - L.24: "Thus, in less developed northwestern China" → "Thus, in this less developed area".

P.12 - L.31: "number" → "few number".

P.12 - L.31: See comment on P.12 - L.10.

P.13 - L.18: In the Conclusion, you could say that this study can pave the way to more detailed and refined analyses based on local data and other sources of information, which represents the next, necessary step to help driving investments in new ski-resorts.

P.13 - L.26: "base" → "based".

P.13 - L.27-28: "the rationality of our suitability evaluation methods was verified based" → "our method to estimate the suitability was evaluated based".

P.14 - L.2: Please consider adding some references on future snow conditions in China.

P.14 - L.3: "may become better" → "may increase".

P.14 - L.4: "become central" → "central".

**Figure 1**: In all figures, you used Rainbow palettes, which are known to have several drawbacks (https://betterfigures.org/2015/06/23/picking-a-colour-scale-for-scientific-graphics/). I strongly suggest you to use different color-bars to improve readability.

**Figure 4**: It looks like there are other regions showing high socioeconomic suitability in Northern and North-Eastern China. Why did you neglect them?

**Figure 6**: You could reverse the y-axis of the color-bar to be consistent with Fig.8. See also comment on P.10 - L.24. Finally, you could maybe swap Fig.6 and Fig.7.

**Figure 8**: Please consider using the same symbols as those used in Fig.7. It seems that several ski areas established before 2012 (red triangles) match recent ski areas (blue circles): did you count the same areas twice? You should also remind that natural and socioeconomic suitabilities have different weight coefficients (0.52 and 0.48, respectively), that is why the shaded regions and the green line do not intercept the x and y axes at the same point. Finally, in the label: "less" → "lower", "dotted" → "dashed".

**Table 3**: Why is the sum of natural suitability coefficients equal to 1.01, and not 1? You should comment on the fact that the total weight of natural suitability (0.52) is higher than that of socioeconomic suitability (0.48). How was this result obtained? See comment on P.8 - L.4.

**Table 4**: "The areas of four zones by different driving factors." → "Surface areas and

driving factors for four zones of suitability."

**Bibliography**

Damm, A., Koeberl, J., and Prettenthaler, F.: Does artificial snow production pay under future climate conditions? - A case study for a vulnerable ski area in Austria, Tourism Management, 43, 8–21, doi:10.1016/j.tourman.2014.01.009, 2014.

Damm, A., Greuell, W., Landgren, O., and Prettenthaler, F.: Impacts of +2°C global warming on winter tourism demand in Europe, Climate Services, 7, 31–46, doi:10.1016/j.cliser.2016.07.003, 2017.

Demiroglu, O. C.: Skiing, climate change, regional development and terrorism: a GIS-based suitability analysis for ski tourism in Turkey. In: Massart, C. (ed) Ski resorts and global challenges, Peter Lang, Bern, in review, 2019.

Demiroglu, O. C., Turp, M. T., Kurnaz, M. L., and Abegg, B.: The Ski Climate Index (SCI): Fuzzification and a Regional Climate Modelling Application for Turkey, Journal of Biometeorology, in review, 2019.

Hanzer, F., Marke, T., and Strasser, U.: Distributed, explicit modeling of technical snow production for a ski area in the Schladming region (Austrian Alps), Cold Regions Science and Technology, 108, 113–124, doi:10.1016/j.coldregions.2014.08.003, 2014.

Spandre, P., Morin, S., Lafaysse, M., George-Marcelpoil, E., François, H., and Lejeune, Y.: Determination of snowmaking efficiency on a ski slope from observations and modelling of snowmaking events and seasonal snow accumulation, Cold Regions Science and Technology, 11, 891–909, doi:10.5194/tc-11-891-2017, 2017.

Spandre, P., François, H., Verfaillie, D., Pons, M., Vernay, M., Lafaysse, M., George, E., and Morin, S.: Winter tourism and climate change in the Pyrenees and the French Alps: relevance of snowmaking as a technical adaptation, The Cryosphere Discuss., doi:10.5194/tc-2018-253, 2018.

Steiger, R.: The impact of climate change on ski season length and snowmaking requirements in Tyrol, Austria, Climate research, 43, 251, doi:10.3354/cr00941, 2010.

Vanat, L.: 2018 International Report on Snow & Mountain Tourism: Overview of the key industry figures for ski resorts, Genève, URL https://www.vanat.ch/RM-world-report-2018.pdf, 2019.

Verfaillie, D., Lafaysse, M., Déqué, M., Eckert, N., Lejeune, Y., and Morin, S.: Multi-component ensembles of future meteorological and natural snow conditions for 1500

m altitude in the Chartreuse mountain range, Northern French Alps, The Cryosphere, 12, 1249–1271, doi:10.5194/tc-12-1249-2018, 2018.

---

## Short Comment (SC1) · 11 Apr 2019

This review commentary will build on the insightful comments of Carlo Maria Carmagnola and not repeat the questions and suggested areas of revision previously identified.

The objective of the analysis is clear and builds on a similar type of analysis in a much smaller study area of the United States that is characterized by very different climateology (particularly natural snow resources).

The approach to include both natural conditions and the socioeconomic factors that can influence the feasibility and competitiveness of ski area operations is essential.

[Figure]

The selection of indicators and how they are operationalized (including data sources) is well explained.

There are two important limitations to the study as currently conducted: First, the equal weighting of the indicators is problematic. The set of indicators are not equal, as some are essential (must be achieved) and others are useful to improve competitiveness or a higher quality ski experience. Consultation with industry stakeholders in China could have been used to determine which indicators are essential to business operations and profitability. Expert weightings could also have been used. For example, without sufficient cold temperatures and water supply for snowmaking, ski operations are not feasible in most of China (which has a dry season in winter and very limited natural snow). Other natural and socio-economic factors cannot overcome the inability to produce and maintain a reliable, quality snowpack at operational depths.

Second, while the range of indicators included is a strength of the study, some of the indicators used are problematic. The study should consult Steiger et al (2017) for a summary of limitations of studies in the literature that result in the mis-represention of climate variability/climate change risks for the ski industry. This study suffers from two of these limitations: (1) the use of inappropriate performance indicators and (2) the lack of an indicator that accurately represents the combined natural-technical snowpack, and therefore does not represent the current operational realities of ski areas in China.

The paper states that, "Therefore, in this study, an SCD [snow cover day] larger than 100 days is taken as the optimal value." Snow cover days are not an suitable indicator of ski seasons. Snow cover is measured as 1 inch/2.5cm and is not sufficient for ski operations, and therefore provides no meaningful information on whether skiable conditions were present on a day with 'snow cover'.

The indicator needed to define a ski season is how many days with sufficient snow depth for ski operations (usually a minimum of 30 cm is used in the literature, but this varies based on terrain). In every regional market in the world, this operational depth

must be calculated as the combined snowpack of natural snow and machine-made snow, because there is no regional market where at least some ski areas / ski terrain utilizes snowmaking. This is particularly the case in China, much of which has a dry winter climate

The study identifies a snow depth threshold that is common in the literature, but provides no measure of how many days this threshold is achieved with natural snow, because "small-scale snow properties (∼1 km) cannot be obtained due to the low resolution of passive microwave products." Furthermore, the study does not physically model the snowpack with snowmaking, but rather uses proxies of potential snowmaking days.

It is not clear what is meant in the statement that, "Therefore, SD is only taken as a reference for the index of snow cover." However, as indicated, snow cover is not a meaningful indicator for ski area operations and cannot be used as a proxy for operational snow depth. Because the study does not provide a robust and meaningful analysis of snow resources (natural or with the additional capacity of snowmaking) that are fundamental to ski operations, publication cannot be recommended.

---

## Author Comment (AC1) · 7 Jun 2019

Daniel Scott

daniel.scott@uwaterloo.ca

This review commentary will build on the insightful comments of Carlo Maria Carmagnola and not repeat the questions and suggested areas of revision previously identified.

The objective of the analysis is clear and builds on a similar type of analysis in a much smaller study area of the United States that is characterized by very different climateology (particularly natural snow resources).

The approach to include both natural conditions and the socioeconomic factors that can influence the feasibility and competitiveness of ski area operations is essential. The selection of indicators and how they are operationalized (including data sources) is well explained.

**Response:** We appreciate Dr. Daniel Scott for his constructive comments on our manuscript.

Before going further, we would like to point out that the ski industry in China is very different from that in major skiing countries (e.g., the United States), not only in terms of climatic conditions but also the development status of the ski market. As described in our manuscript, China's ski industry is still at the stage of rapid development. Due to the lack of strategic plan and effective regulation, the chaotic market has caused serious environmental problems and waste, which hampers the sustainable development of the ski industry. Therefore, we would like to restate the significance of our work considering the results that provide timely advice and guidance to sustainable development of the ski market based on natural and socioeconomic conditions in China.

In summary, this study was designed to 1) provide a scientific metric for future development of ski market at the national scale and 2) evaluate the current situation of the ski resorts, which is definitely lacking and most-needed in the nation. Additionally, a more sophisticated and object-oriented method should be further developed when considering a specific resort (such as the snowmaking), which, indeed, can be derived and modified based on the current method.

We have carefully addressed all the issues raised by the Reviewer and modified our manuscript accordingly. Detailed responses (marked in blue font) are summarized in the following sections with the original comments (marked in black font). The revised manuscript is attached with changes marked in red font.

There are two important limitations to the study as currently conducted: First, the equal weighting of the indicators is problematic. The set of indicators are not equal, as some are essential (must be achieved)

and others are useful to improve competitiveness or a higher quality ski experience. Consultation with industry stakeholders in China could have been used to determine which indicators are essential to business operations and profitability. Expert weightings could also have been used. For example, without sufficient cold temperatures and water supply for snowmaking, ski operations are not feasible in most of China (which has a dry season in winter and very limited natural snow). Other natural and socio-economic factors cannot overcome the inability to produce and maintain a reliable, quality snowpack at operational depths.

**Response:**

(1) First, we would like to address the question regarding the weights of indicators. We agree with the Reviewer that the weights of indicators are not equal. Therefore, in this study, a common method based on entropy weight theory was used (Bian et al., 2018; Bednarik et al., 2010; Srdjevic, Medeiros, and Faria, 2004; Tang, 2015) to determine the weights based on both statistics and expert knowledge. Indeed, calculations of the index weights are divided into two groups: One is determined by the knowledge and experience of experts or individuals (including the stakeholders mentioned by the Reviewer), named the subjective weight; the other is based on statistical properties and measured data, named the objective weight. The entropy weight method belongs to the latter as one of the objective methods to determine weight weighting coefficients. It has been commonly used in the fields of sustainable development evaluation and social economy (Vranešević et al., 2016). The weight coefficients of each indicator were calculated based on the sample of 116 existing ski areas established before 2012 (see Table 3). As the reviewer mentioned, the evaluation results obtained by the entropy weight method, relying on the objective data, may be slightly deviated from the expert's understanding. However, it is commonly subjective to determine the weights according to the expert experiences. In a word, we believe that both methods have their values even we chose to use the objective method for our weights in this study. Nevertheless, we incorporated government report and field surveys of 35 ski resorts including questionnaires and interviews with managers and staff, which, though, not used in the weights calculation, but could be served as complementary information to evaluate our results. In the revised manuscript, we added more explanations of the entropy weight method as follows (Page 9, Line 28 to Page 10, Line 3):

"The weight coefficients were calculated by an objective method based on the theory of information entropy, which has been widely employed for the determination of weights of evaluating indicators. (Bian et al., 2018; Bednarik et al., 2010; Srdjevic, Medeiros, and Faria, 2004; Vranešević et al., 2016). The concept of information entropy originally came from thermodynamics and indicates the extent of disorder in the system status (Bednarik et al., 2010). Generally, if the dispersion of data is high, the value of information entropy is low, which means that more information will be provided. Correspondingly, the greater the influence of the index on the evaluation, the higher its weight (Tang, 2014). Therefore, the entropy weight method can be used to calculate the objective weights of the index

system and avoid bias caused by subjectivity to a certain extent (Pourghasemi, Mohammady, and Pradhan, 2012)."

According to the results of the weight coefficients (Table 3), we also added the following sentences in the end of Sect. 2.3 (Page 10, Line 26-28):

"The results show that the total weight of natural suitability (0.52) is higher than that of socioeconomic suitability (0.48), which indicates that natural conditions have a greater impact on the development of ski areas than socioeconomic conditions."

Table 3. Weight coefficients for the evaluation indexes.

| Indexes | Weight coefficients |
| --- | --- |
| Natural suitability | 0.52 |
| Snow cover | 0.36 |
| Air temperature | 0.19 |
| Topographic conditions | 0.32 |
| Water resources | 0.04 |
| Vegetation | 0.09 |
| Socioeconomic suitability | 0.48 |
| Economic conditions | 0.37 |
| Distance to a city | 0.29 |
| Accessibility of transportation | 0.18 |
| Distance to a tourist attraction | 0.16 |

(2) Then we would like to discuss our results in China compared with other developed countries. In European and American countries, people pursue high-quality skiing experience, so natural factors are crucial to the operation of ski resorts. The Reviewer may be surprised by the results of weight coefficients in Table 3, which show such high weights of socioeconomic indicators. To help explain the results, we plot spatial distributions of natural suitability and integrated suitability in the existing ski areas. As shown in Fig. S1 (supplementary information), in the eastern and southern of China, natural conditions are relatively modest. A large number of small-size ski areas are located in such areas with better socioeconomic conditions. The results indicate that, in the developing country like China, socioeconomic conditions are still very important factors for the operation of a ski area at present.

[Figure]

Figure S1. The distribution of existing ski areas in China. (a) Natural suitability map; (b) Integrated suitability map.

However, with social and economic development, these ski areas are threatened with closure, causing further environmental problems. Therefore, when making an investment on a new ski area, not only the integrated suitability but also the driving factors (see Sect. 3.3) should be considered. In the discussion section, we have also pointed this out as:

"In total, 84.23% of the ski areas were located in areas with an integrated suitability value greater than 0.5, while 15.77% of the ski areas were located in areas with integrated suitability values less than 0.5, which are almost all distributed in socioeconomic-driven areas. Notably, the ski areas in socioeconomic-driven areas are more prone to environmental problems due to the decreased natural suitability, and these enterprises may soon face the challenges of dismal prospects." (Page 13, Line 26-30)

"In socioeconomic-driven areas, the ski area markets are determined by local visitors. According to the situation of local socioeconomic development, an appropriate number of small ski areas are advised to be built for recreational sports to expand the influence of snow sports. However, a large number of small-size ski areas with poor facilities have been established in the economically driven areas, and these ski areas are unsafe and provide skiers with low-quality ski experiences. Therefore, in socioeconomic-driven areas, the number of ski areas should be limited, and enterprises should enhance their competitiveness by improving the quality of the ski area." (Page 14, Line 25-30)

(3) To further illustrate the reliability of the results, the following figures are presented as supplementary information that were not included in the manuscript. As we can see, the higher the grade of the ski area is, the better its locational integrated suitability (Fig. S2). Further, we also analyzed the locational integrated suitability and the ski season lengths of ski areas. As shown in Fig. S3, the ski season length varies widely from 60 to 182 days. With the increase of locational suitability, the snow season lengths in ski areas is getting longer. The low determination coefficient between ski season length and integrated suitability is due to the fact that some ski areas are still operated even under the condition of poor snow quality.

[Figure]

Figure S2. Locational integrated suitability versus the grades (ski areas that were investigated).

[Figure]

Figure S3. Locational integrated suitability versus the ski season lengths (ski areas that were investigated).

Second, while the range of indicators included is a strength of the study, some of the indicators used are problematic. The study should consult Steiger et al (2017) for a summary of limitations of studies in the literature that result in the mis-represention of climate variability/climate change risks for the ski industry. This study suffers from two of these limitations: (1) the use of inappropriate performance indicators and (2) the lack of an indicator that accurately represents the combined natural-technical snowpack, and therefore does not represent the current operational realities of ski areas in China.

**Response:** We appreciate the Reviewer for recommending Steiger et al. (2017). We carefully read the paper and found that this work is very interesting, which provided a critical review of studies on the risk of climate change on ski tourism. However, in this study, we mainly analyzed the current suitability for development of ski areas in term of natural and socioeconomic conditions, not considering the potential influence of climate/environmental changes. Our future research will focus on the impact of climate change and socioeconomic development on ski market, which has been discussed in the revised manuscript (Page 16, Line15-24):

"The results of the weight coefficients indicate that snow resources, air temperature and topographic conditions are major natural factors that influence ski area development. In the context of global warming, it is necessary to evaluate the vulnerability of a ski area to future climate change (Steiger et al., 2017). With the increase in winter temperatures and the decrease in snowfall in the next few decades (Ji and Kang, 2012; Wang and Wang, 2012), the natural suitability values for southern and eastern China will decrease, and small midlatitude and low-elevation ski areas will be the first to close due to poor snow conditions (Bark and Colby, 2010; Gilaberte-Búrdalo et al., 2017). Additionally, with social and economic development, people's living standards will greatly improve. Thus, the socioeconomic suitability in northwestern and northeastern China may increase. As a result, northwestern and northeastern China may become popular markets and central places for ski tourism. To more thoroughly study the future of ski tourism in China, future research is needed to evaluate the locational suitability of ski areas in relation to climate change and socioeconomic development."

The paper states that, "Therefore, in this study, an SCD [snow cover day] larger than 100 days is taken as the optimal value." Snow cover days are not an suitable indicator of ski seasons. Snow cover is measured as 1 inch/2.5cm and is not sufficient for ski operations, and therefore provides no meaningful information on whether skiable conditions were present on a day with 'snow cover'.

The indicator needed to define a ski season is how many days with sufficient snow depth for ski operations (usually a minimum of 30 cm is used in the literature, but this varies based on terrain). In every regional market in the world, this operational depth must be calculated as the combined snowpack of natural snow and machine-made snow, because there is no regional market where at least some ski areas / ski terrain utilizes snowmaking. This is particularly the case in China, much of which has a dry winter climate.

The study identifies a snow depth threshold that is common in the literature, but provides no measure of how many days this threshold is achieved with natural snow, because "small-scale snow properties (∼1 km) cannot be obtained due to the low resolution of passive microwave products." Furthermore, the study does not physically model the snowpack with snowmaking, but rather uses proxies of potential snowmaking days.

It is not clear what is meant in the statement that, "Therefore, SD is only taken as a reference for the index of snow cover." However, as indicated, snow cover is not a meaningful indicator for ski area

operations and cannot be used as a proxy for operational snow depth. Because the study does not provide a robust and meaningful analysis of snow resources (natural or with the additional capacity of snowmaking) that are fundamental to ski operations, publication cannot be recommended.

**Response:** We appreciate the Reviewer's feedback. Overall, this study was designed to provide a scientific metric for the development of ski market at the national scale. Therefore, the choice of indicators is useful for the evaluation in large scale. We considered snow conditions from two aspects of natural snow and machine-made snow, in which the machine-made snow was reflected by the air temperature conditions. We agree that there are limitations in the processing of snow cover index and the treatment of machine-made snow. Snow model, which integrates natural and machine-made snow, can better reflect local snow conditions for specific ski area (Scott, McBoyle, and Mills, 2003; Hennessy et al., 2008; Spandre et al., 2017). But it is difficult to apply the site-scale model to a large scale. For future study, we plan to select several typical ski areas to develop a more sophisticated and object-oriented method.

a. First, we would like to respond to the comments on performance indicators. On the one hand, as suggested by the Reviewer, snow cover day may not be a suitable indicator on skiable conditions. In this study, snow cover day is considered as an indicator of climate suitability (temperature and precipitation) and the measure of aesthetic value. On the other hand, since the areas in China with natural snow depths greater than 30 cm are very few, we did not used this measure. Instead, the average snow depth (SD) was considered in this study.

In the revised manuscript, the following two sentences:

"Therefore, SD is only taken as a reference for the index of snow cover." and "small-scale snow properties (∼1 km) cannot be obtained due to the low resolution of passive microwave products." have been rephrased as follows (Page 4, Line 17-28):

"Natural snow cover is a crucial resource for ski areas. Skiers may cancel trips when there are poor snow conditions (Scott et al., 2003; Steiger and Abegg, 2018). Tervo (2008) analyzed the viability of nature-based winter tourism enterprises and declared that 90–120 skiable days are adequate for making a profit. In fact, a ski area is profitable if the snow reliability period is greater than 100 days per season, which is known as the 100-day rule and is the most common indicator of snow reliability (Steiger, 2012). Additionally, Scott, McBoyle, and Mills (2003) defined a skiable day as a day with a snow depth greater than 30 cm on ski runs. However, the snow depth in China is much lower than that in North America and Europe, and the areas with natural snow depths greater than 30 cm are extremely rare (Mudryk et al., 2015). Therefore, we did not use the indicator of the number of days with a natural snow depth greater than 30 cm. As a supplement for the snow depth on ski runs, the average snow depth (SD) during a ski season was considered in this study. In addition, the number of snow cover days (SCD), which is total number of days (can be discontinuous) with snow cover in an area during the ski season, was considered as an indicator of

climate suitability (temperature and precipitation) and a measure of the aesthetic value of a site given that many ski areas have snow cover on only the ski runs."

b. Second, we agree with the Reviewer, that the conditions of snowmaking are the important factor for ski destination choice. In the revised manuscript, the following sentence has been added to the end of the section of **Snow cover** (also was suggested by Reviewer #1):

"This index was for natural snow only, and machine-made snow was taken into account by the index of air temperature." (Page 5, Line 9-10)

We reclassified the daily mean temperature into 11 regimes. Among them, the air temperature between -2 ℃ and -5 ℃ are were taken as optimal conditions in regard to efficient snowmaking.

We added some more explanation in the section of **Air temperature** (also was suggested by Reviewer #1):

"The 11 temperature regimes and their corresponding scores were designed as a trade-off between the cold temperatures needed to preserve the snowpack and to produce machine-made snow and the warm temperatures needed by skiers." (Page 5, Line 28-30)

In this study, we did not physically model the snowpack with snowmaking, but used proxies of potential snowmaking days. Therefore, we added the relevant discussions in a new section:

"The first limitation of our study is related to machine-made snow. For the index of snow cover, only natural snow has been considered. As mentioned in Sect. 2.1.1, machine-made snow was considered in the index of air temperature, which may have imperfectly represented the snowmaking conditions. The exact number of skiable days cannot be captured by using our method. Some studies have focused on modeling machine-made snow processes, and these models have been used to calculate the length of the ski season at specific ski destinations (Scott, McBoyle, and Mills, 2003; Hennessy et al., 2008; Spandre et al., 2017). The main barrier that arises when addressing snow models over large-scale regions is associated with the difficulty in obtaining data with high spatial and temporal resolution. It is also difficult to apply the site-scale model in a large scale. However, since the aim of this study was to provide the guidelines for the ski market at the national scale rather than define whether a specific resort will be viable, we believe that using the air temperature to reflect the snowmaking conditions is acceptable over large-scale regions." (Page 15, Line 5-14)

"The method of this work attempts to identify the suitability patterns of the main ski areas in China, but a more detailed and refined analysis based on local data will be necessary before deciding to invest (or not) in a new ski area. Based on previous studies, machine-made snow, air temperature and wind should be addressed in future studies on specific ski areas." (Page 15, Line 28-30)

The associated discussion also was added in the conclusion section:

"This study can pave the way for more detailed and refined analyses based on local data and other sources of information, which represents the next necessary step to promote investments in new ski areas." (Page 16, Line 12-14)

**References**

Bednarik, M., Magulova, B., Matys, M., and Marschalko, M.: Landslide susceptibility assessment of the Kralovany–Liptovsky Mikulaš railway case study, Phys. Chem. Earth, 35, 162–171, doi.org/10.1016/j.pce.2009.12.002, 2010.

Bian, Z., Xu, Z., Xiao, L., Dong, H., and Xu, Q.: Selection of optimal access point for offshore wind farm based on multi-objective decision making, Int. J. Electr. Power Energy Syst., 103, 43–49, doi:10.1016/j.ijepes.2018.05.025, 2018.

Hennessy, K. J., Whetton, P. H., Walsh, K., Smith, I. N., Bathols, J. M., Hutchinson, M., and Sharples, J.: Climate change effects on snow conditions in mainland Australia and adaptation at ski resorts through snowmaking, Clim. Res., 35, 255–270, doi:10.3354/cr00706, 2008.

Srdjevic, B., Medeiros, Y. D. P., and Faria, A. S.: An Objective Multi-Criteria Evaluation of Water Management Scenarios. Water Resour. Manag., 18, 35–54, doi:10.1023/b:warm.0000015348.88832.52, 2004.

Scott, D., McBoyle, G., and Mills, B.: Climate change and the skiing industry in southern Ontario (Canada): Exploring the importance of snowmaking as a technical adaptation, Clim. Res., 23, 171–181, doi:10.3354/cr023171, 2003.

Spandre, P., Morin, S., Lafaysse, M., George-Marcelpoil, E., Fran¸cois, H., and Lejeune, Y.: Determination of snowmaking efficiency on a ski slope from observations and modelling of snowmaking events and seasonal snow accumulation, The cryosphere, 11, 891-909, doi:10.5194/tc-11-891-2017, 2017.

Steiger, R., Scott, D., Abegg, B., Pons, M., and Aall, C.: A critical review of climate change risk for ski tourism, Curr. Issues Tour., 1-37, doi.org/10.1080/13683500.2017.1410110, 2017.

Tang, Z.: An integrated approach to evaluating the coupling coordination between tourism and the environment, Tour. Manag., 46, 11–19, doi:10.1016/j.tourman.2014.06.001, 2015.

Vranešević, M., Belić, S., Kolaković, S., Kadović, R., and Bezdan, A.: Estimating Suitability of Localities for Biotechnical Measures on Drainage System Application in Vojvodina, Irrig. Drain., 66, 129–140, doi:10.1002/ird.2024, 2016.

---

## Author Comment (AC2) · 7 Jun 2019

Carlo Maria Carmagnola

carlo.carmagnola@meteo.fr

**Summary:**

This study presents a method to evaluate the suitability of existing and future ski areas in China based on a combination of natural and socioeconomic conditions. Blending geographic information system spatial analysis, remotely-sensed observations, online and field data, the authors were able to define an integrated index to identify the best suited locations for ski-resorts in the country. This index was evaluated using the locations of already existing resorts and more detailed information from a field survey of a panel of 35 resorts. Results show that 92% of the existing ski-resorts are currently located in areas with medium-to-high suitability indexes. As for future ski-resorts, this analysis framework could help driving new investments in the ski market at the national scale, by defining objective indicators to identify regions potentially interesting for developing ski-related activities. To this aim, the authors also suggest some broad strategies to guide the next steps of ski tourism industry in China.

**Response:** We are grateful to Dr. Carlo Maria Carmagnola for his positive and constructive comments, which provided tremendous help for improving our manuscript. We have carefully addressed all the issues raised by him and modified our manuscript accordingly. Detailed responses (marked in blue font) are summarized in the following sections with the original comments (marked in black font). The revised manuscript is attached with changes marked in red font.

The main novelty of this work lies in the method used for aggregating the different natural (snow cover, air temperature, topography, groundwater, vegetation) and socioeconomic (welfare, accessibility of transportation, distance to tourist attractions, distance to cities) conditions. Applying a linear weighting method allowed the authors to compute the relative contributions of each condition to the integrated suitability index. However, this part of the work also represents the weakest section of the paper in its current form. Indeed, the approach followed to compute and then aggregate the weight coefficients starting from a data sample of 128 ski-resorts is not presented clearly. In particular, it appears that the same dataset has been used for informing the aggregation method and to evaluate its results. Moreover, and perhaps more importantly, the computation of individual indexes, namely those for snow cover and air temperature, raises some concerns. In this regard, for example, is it not clear how the snow depth and snow cover days have been estimated and there seems to be some confusion between air temperature and wet bulb temperature, which is the main parameter for snowmaking. More generally,

I would not overstate the importance of the main results of this study for a direct application. Even if the method can be seen as a first attempt to identify the main ski area's suitability pattens in China, a more detailed and refined analysis based on local data will be necessary before deciding to invest (or not) in a new ski-resort. In other words, the results of this study provide interesting guidelines for the ski market at the national scale, but do not tell whether a specific resort will be, in fact, viable.

**Response:** We thank the Reviewer for all the suggestions and understand his concerns. In summary, this study was designed to 1) provide a scientific metric for future development of ski market at the national scale and 2) evaluate the current situation of the ski resorts, which is definitely lacking and most-needed in the nation. We agree with the reviewer that a more sophisticated and object-oriented method should be further developed when considering a specific resort, which, indeed, can be derived and modified based on the current method. We acknowledge the Reviewer for his vision and added the associated discussions in the discussion section, as well as the conclusion section. In the following sections, we try to address the issues point-by-point with changes in the manuscript.

a. However, this part of the work also represents the weakest section of the paper in its current form. Indeed, the approach followed to compute and then aggregate the weight coefficients starting from a data sample of 128 ski-resorts is not presented clearly.

   **Response:** We apologize for the confusing description of the linear weighting method. For clarity, we modified Sect. 2.3 with additional details reference to comment #41 (Page 9, Line 13 to Page 10, Line 23):

   "In this study, the linear weighting method was used for synthetic evaluation. The natural suitability for ski area development (NS) is expressed as

   $$NS = \sum_{i=1}^{i} W_i NC_i \tag{1}$$

   where $i$ is the number of natural condition indexes ($i$ = 1, 2,…, 5), $W_i$ is the weighting factor that represents the importance of the natural condition indexes, and $NC_i$ represents the natural index. The socioeconomic suitability for ski area development ($SS$) is expressed as

   $$SS = \sum_{j=1}^{j} W_j SC_j \tag{2}$$

   where $j$ is the number of the socioeconomic condition indexes ($j$ = 1, 2,…, 4), $W_j$ is the weighting factor that represents the importance of the socioeconomic indexes, and $SC_j$ is the socioeconomic index.

   The integrated suitability for ski area development ($IS$) is expressed as

   $$IS = \sum_{f=1}^{f} W_f C_f \tag{3}$$

   where $f$ is the number of indexes ($f$ = 1, 2), $W_f$ is the weight coefficient, $C_f$ is the index of $NS$ or the index of $SS$.

   The weight coefficients in Eqs. (1-3) ($W_i$, $W_j$ and $W_f$) are defined using the entropy method introduced as follows. Finally, $NS$, $SS$ and $IS$ are rescaled within the range of 0-1.

The 116 existing ski areas established before 2012 were used as samples to compute the weight coefficients by extracting all index values for the ski area location. The weight coefficients were calculated by an objective method based on the theory of information entropy, which has been widely employed for the determination of weights of evaluating indicators. (Bian et al., 2018; Bednarik et al., 2010; Srdjevic, Medeiros, and Faria, 2004; Vranešević et al., 2016). The concept of information entropy originally came from thermodynamics and indicates the extent of disorder in the system status (Bednarik et al., 2010). Generally, if the dispersion of data is high, the value of information entropy is low, which means that more information will be provided. Correspondingly, the greater the influence of the index on the evaluation, the higher its weight (Tang, 2014). Therefore, the entropy weight method can be used to calculate the objective weights of the index system and avoid bias caused by subjectivity to a certain extent (Pourghasemi, Mohammady, and Pradhan, 2012).

It is supposed that there are $a$ optional schemes, each with $b$ evaluating indexes. The data matrix is established as follows:

$$X = \begin{bmatrix} x_{11} & \cdots & x_{1a} \\ \vdots & \ddots & \vdots \\ x_{b1} & \cdots & x_{ba} \end{bmatrix}_{a \times b} \tag{4}$$

where $x_{nm}$ is the value of the $m^{\text{th}}$ index in the $n^{\text{th}}$ scheme, $n = 1, 2,..., a$; $m = 1, 2,..., b$. $a$ is the number of ski areas established before 2012 ($n = 116$), $b$ is the number of indexes in Eqs. (1-3) ($i$, $j$ and $f$).

it is noted that the indexes are not measured on the same scale, and they have different dimensions. If the analysis is performed directly with the original index values, the difference in different quantity grades in X may produce inaccurate results in the decision-making process. Therefore, normalization ($Z_{nm}$) is performed to make the indexes dimensionless and mutually comparable, which is expressed as

$$Z_{nm} = \frac{x_{m,max} - x_{nm}}{x_{m,max} - x_{m,min}} \tag{5}$$

where

$$x_{m,max} = max\{x_{1m}, \ldots x_{am}\} \tag{6}$$

$$x_{m,min} = min\{x_{1m}, \ldots x_{am}\} \tag{7}$$

The information entropy $e_m$ of the $m^{\text{th}}$ index is defined as

$$e_m = -k \sum_{n=1}^{a} P_{nm} \ln(P_{nm}) \tag{8}$$

where

$$k = \frac{1}{\ln a} \tag{9}$$

$$P_{nm} = \frac{Z_{nm}}{\sum_{n=1}^{a} Z_{nm}} \tag{10}$$

if $P_{nm} = 0$, $P_{nm} \ln(P_{nm}) = 0$.

The entropy weight $w_m$ of the $m^{\text{th}}$ index is expressed as

$$w_m = \frac{1 - e_m}{\sum_{m=1}^{b}(1 - e_m)} \tag{11}"$$

b.  In particular, it appears that the same dataset has been used for informing the aggregation method and to evaluate its results.

**Response**: Sorry about the confusion. We divided the ski areas into three categories with different purposes, and used different datasets to inform the aggregation method and to validate its results. We clarified it by rewriting Sect. 2.2 as (Page 8, Line 23 to Page 9, Line 11):

"Although there were indeed 742 ski areas in China according to the International Report on Snow and Mountain Tourism in 2018 (Vanat, 2019), only 598 ski areas' information was manually collected from an online map (https://maps.baidu.com), because limited information was available online for lower-class ski areas. The collected ski areas were divided into three categories:

(1) 116 ski areas before 2012 as sample data: It is often believed that successful and suitable ski areas could operate for long periods of time. We then considered ski areas that had been in operation for more than 6 years as successful and suitable ski areas. As such, the location information for 116 operating ski areas established before 2012 (mainly distributed in northern, northeastern and northwestern China, Fig. 7) was categorized as suitable sample data, which were used to calculate the weight coefficients for the different indexes.

(2) 35 ski areas during field surveys in 2018 for validation: To validate the results, field surveys on ski areas were conducted during the 2018 ski season over 13 provinces, spatially representing different natural and socioeconomic conditions (see Fig. 7). The managers or staff of 35 ski areas were interviewed with a questionnaire and through conversations, mainly involving location information, ski season length, the quality of the ski area, its operating state, and its market competition situation. The questions concerning the quality of the ski area were also designed to collect the characteristics of the ski area, including the number and slope of ski runs, equipment, facilities, and accessibility of transportation. According to the Rank Division for the Quality of Mountain Ski Resorts released by the China National Tourism Administration in 2014, the quality of ski areas was evaluated by five grades from 1S to 5S, representing the lowest- to the highest-grade standard. This procedure involved comprehensive assessment integrating 10 aspects, such as equipment, facilities, accessibility of transportation, and the reception scale. According to the feedback from our field surveys, ski areas were roughly ranked into different grades, except for the 8 ski areas whose grades were already defined by the Xinjiang Government Tourist Office.

(3) 447 ski areas for evaluation: Additionally, the locational suitability of the 447 ski areas established after 2012 was evaluated using the aggregation method."

We used field survey information to validate the evaluation method, and then the method were used to evaluate the locational suitability of 447 ski areas established after 2012. Correspondingly,

descriptions of the validation and evaluation in Sect. 4.1 were modified (Page 13, Line 6-30). We also reprocessed Fig. 7 and Fig. 8:

[revised manuscript text omitted]

c.  Moreover, and perhaps more importantly, the computation of individual indexes, namely those for snow cover and air temperature, raises some concerns. In this regard, for example, is it not clear how the snow depth and snow cover days have been estimated and there seems to be some confusion between air temperature and wet bulb temperature, which is the main parameter for snowmaking.

**Response**: We agree with the Reviewer that there were some issues with the computation of individual indexes. Refer to the comment #22 and #23, we modified the sections of snow cover and air temperature, respectively, to explain how we estimated the snow cover index, as well as the air temperature index.

**Snow cover:**

(1) In this study, the ski season was defined as the period from November 1 to March 31 (Page 4, Line 4-6):

"Regions where the elevation is higher than 4000 m are unsuitable for skiing due to the lack of oxygen, as are the areas where the maximum air temperature is higher than 10 ℃ for more than 90 days during the ski season (November 1 to March 31)."

For clarity, we changed "ski season" to "skiable days" in the following sentence (Page 4, Line 18-19):

"Tervo (2008) analyzed the viability of nature-based winter tourism enterprises and declared that 90–120 skiable days are adequate for making a profit."

(2) The snow cover day (SCD) is primarily acquired from optical remote sensing snow cover product (the moderate-resolution imaging spectroradiometer, MODIS) rather than from snow depth images. In the revised manuscript, we rephrased the following sentence (Page 4, Line 29-31):

"With a spatial resolution of 500 m, version 4 of the moderate-resolution imaging spectroradiometer (MODIS) snow cover products MOD10A1 and MYD10A1were used in this study to obtain SCD, and these products were downloaded from the National Snow Ice Data Center (NSIDC, https://nsidc.org/)."

(3) The sentence that "snow depth is only taken as a reference for the index of snow cover." have been rephrased as follows (Page 4, Line 22-25):

"However, the snow depth in China is much lower than that in North America and Europe, and the areas with natural snow depths greater than 30 cm are extremely rare (Mudryk et al., 2015). Therefore, we did not use the indicator of the number of days with a natural snow depth greater than 30 cm. As a supplement for the snow depth on ski runs, the average snow depth (SD) during a ski season was considered in this study."

(4) When applying the 100-day-rule, the non-consecutive days have been considered. We added more explanations as (Page 4, Line 25-28):

"In addition, the number of snow cover days (SCD), which is total number of days (can be discontinuous) with snow cover in an area during the ski season, was considered as an indicator of climate suitability (temperature and precipitation) and a measure of the aesthetic value of a site given that many ski areas have snow cover on only the ski runs."

(5) The moderate-resolution imaging spectroradiometer (MODIS) snow cover products (MOD10A1 and MYD10A1, version 4) have been used in this study. The Normalized Difference Snow Index (NDSI) calculated by using the fourth and sixth bands was adopted in MODIS snow cover images to distinguish snow. We revised the sentences (Page 4, Line 29 to Page 5, Line 2):

"With a spatial resolution of 500 m, version 4 of the moderate-resolution imaging spectroradiometer (MODIS) snow cover products MOD10A1 and MYD10A1 were used in this study to obtain SCD, and these products were downloaded from the National Snow Ice Data Center (NSIDC, https://nsidc.org/). The normalized difference snow index (NDSI) was adopted to distinguish snow in the MODIS snow cover images. The NDSI was calculated using the fourth and sixth bands of MODIS, and a pixel was defined as snow if the NDSI value was greater than 0.4. Further details on the MODIS snow cover products can be found in Hall et al. (2002)."

(6) As for the normalization of SD and SCD indexes, we added explanations as (Page 5, Line 7-8):

"An SCD larger than 100 days and SD greater than 30 cm were taken as the optimal conditions for normalization."

(7) As suggested by the Reviewer, we added a short statement to the end of this section (Page 5, Line 9-10):

"This index was for natural snow only, and machine-made snow was taken into account by the index of air temperature."

**Air temperature:**

(1) To explain more clearly how we have computed daily mean and maximum air temperatures starting from a dataset of daily observations, we inserted the following sentences (Page 5, Line 18-21):

"The gridded daily observation dataset includes two variables: daily mean air temperature and daily maximum air temperature. The daily mean air temperature was the average value of the dry-bulb temperature at 02:00, 8:00, 14:00 and 20:00 (local time). The daily maximum air temperature was measured using a maximum thermometer."

(2) The definition of the high-temperature regions has been rephrased as follows (Page 5, Line 24-26):

"In the southern China, many small ski areas can be profitable during a ski season with 60 skiable days. Therefore, the high-temperature region was defined as the area with more than 90 noncontinuous days with maximum air temperatures greater than 10 ℃ during a ski season (151 days)."

(3) The dry-bulb temperature data have been used in this study, and please see the previous reply to **Air temperature** (1). In addition, the relevant explanation that why we did not use the wet-bulb temperature has been added in Sect. 4.3 (Page 15, Line 15-19):

"The second limitation stems from the air temperature data. Subzero wet-bulb temperatures are needed for snowmaking (Hennessy et al., 2008). Unfortunately, meteorological stations do not provide wet-bulb temperature data. As a proxy, the dry-bulb temperature is also widely used in machine-made snow models (Scott, McBoyle, and Mills, 2003; Steiger, 2010). Therefore, in this study, the dry-bulb temperature data were used to generate the index of air temperature, which reflects the snowmaking conditions and tourist comfort."

(4) The Reviewer makes a good suggestion for the temperature regimes. We added the explanation as follows (Page 5, Line 28-30):

"The 11 temperature regimes and their corresponding scores were designed as a trade-off between the cold temperatures needed to preserve the snowpack and to produce machine-made snow and the warm temperatures needed by skiers."

(5) Finally, the effect of wind has been discussed in Sect. 4.3 (Page 15, Line 20-27):

"Another limitation is associated with wind, which has a significant impact on the efficiency of snowmaking and the attractivity of a site (Spandre et al., 2017). Wind speeds higher than 6 m s-1 have a direct physical or mechanical effect on tourists (Freitas, 2005). Daily wind speed data are available from the meteorological stations that are generally located around a city, while most ski areas are built in the mountains far from cities. Hence, the relatively sparse meteorological stations across the study area did not allow for the creation of reliable daily gridded data. In addition, wind is considered one of the most difficult meteorological variables to model due to its dependence on the specific characteristics of any given location, such as topography and surface roughness (Morales, Lang, and Mattar, 2012). The local micrometeorology at a specific location also cannot be captured using existing reanalysis wind speed data with low resolution."

d. More generally, I would not overstate the importance of the main results of this study for a direct application. Even if the method can be seen as a first attempt to identify the main ski area's suitability pattens in China, a more detailed and refined analysis based on local data will be necessary before deciding to invest (or not) in a new ski-resort. In other words, the results of this

study provide interesting guidelines for the ski market at the national scale, but do not tell whether a specific resort will be, in fact, viable.

**Response**: To develop new businesses in a developing country (e.g., China) while maintain environment sustainability is challenging. Our research aims to provide guidelines to the ski market in China in a perfect time when the Winter Olympics is just around the corner. However, as noted by the Reviewer, this work has its limitations and space for future work. In the revised manuscript, we added relevant discussions in a new Sect. 4.3 as follows (Page 15, Line 4-30):

"**4.3 Limitations of this work**

The first limitation of our study is related to machine-made snow. For the index of snow cover, only natural snow has been considered. As mentioned in Sect. 2.1.1, machine-made snow was considered in the index of air temperature, which may have imperfectly represented the snowmaking conditions. The exact number of skiable days cannot be captured by using our method. Some studies have focused on modeling machine-made snow processes, and these models have been used to calculate the length of the ski season at specific ski destinations (Scott, McBoyle, and Mills, 2003; Hennessy et al., 2008; Spandre et al., 2017). The main barrier that arises when addressing snow models over large-scale regions is associated with the difficulty in obtaining data with high spatial and temporal resolution. It is also difficult to apply the site-scale model in a large scale. However, since the aim of this study was to provide the guidelines for the ski market at the national scale rather than define whether a specific resort will be viable, we believe that using the air temperature to reflect the snowmaking conditions is acceptable over large-scale regions.

The second limitation stems from the air temperature data. Subzero wet-bulb temperatures are needed for snowmaking (Hennessy et al., 2008). Unfortunately, meteorological stations do not provide wet-bulb temperature data. As a proxy, the dry-bulb temperature is also widely used in machine-made snow models (Scott, McBoyle, and Mills, 2003; Steiger, 2010). Therefore, in this study, the dry-bulb temperature data were used to generate the index of air temperature, which reflects the snowmaking conditions and tourist comfort.

Another limitation is associated with wind, which has a significant impact on the efficiency of snowmaking and the attractivity of a site (Spandre et al., 2017). Wind speeds higher than 6 m s-1 have a direct physical or mechanical effect on tourists (Freitas, 2005). Daily wind speed data are available from the meteorological stations that are generally located around a city, while most ski areas are built in the mountains far from cities. Hence, the relatively sparse meteorological stations across the study area did not allow for the creation of reliable daily gridded data. In addition, wind is considered one of the most difficult meteorological variables to model due to its dependence on the specific characteristics of any given location, such as topography and surface roughness (Morales, Lang, and Mattar, 2012). The local micrometeorology at a specific location also cannot be captured using existing reanalysis wind speed data with low resolution.

The method of this work attempts to identify the suitability patterns of the main ski areas in China, but a more detailed and refined analysis based on local data will be necessary before deciding to invest (or not) in a new ski area. Based on previous studies, machine-made snow, air temperature and wind should be addressed in future studies on specific ski areas."

The associated discussion also was added in the Conclusion section (Page 16, Line 12-14):

"This study can pave the way for more detailed and refined analyses based on local data and other sources of information, which represents the next necessary step to promote investments in new ski areas."

Overall the paper is easy to follow and well written, beside some small mistakes and sentences that need to be rephrased. In light of the novel approach and the great effort put by the authors in combining and homogenizing data coming from various sources in a consistent and coherent way, I support publication in The Cryosphere. However, several points should be considered to improve the presentation and some major issues should be addressed before publication can be recommended.

**Response:** We thank the Reviewer for his thorough review. For our specific responses please see below.

**Detailed Comments:**

1) P.1 - L.19: "To evaluate" → "Therefore, evaluating".

**Response:** We replaced "To evaluate" by "Therefore, evaluating".

2) P.1 - L.20: "has since become" → "has become".

**Response:** We replaced "has since become" by "has become".

3) P.1 - L.21: "using linear" → "using a linear".

**Response:** Corrected.

4) P.1 - L.22: "information systems" → "information system".

**Response:** Corrected.

5) P.1 - L.25-28: I would reverse the order of two sentences and rephrase them as follows: "As such, a metrics ranging from 0 to 1 considering both natural and socioeconomic conditions is used to define a suitability threshold for each candidate region for ski area development. A ski area is considered to be a dismal prospect when the locational integrated index is less than 0.5. The results show that 92% of existing ski areas are located in areas with an integrated index greater than 0.5".

**Response:** Corrected as suggested.

6) P.2 - L.19: "freshwater resource" → "resource".

**Response:** We replaced "freshwater resource" with "resource".

7) P.2 - L.22: On the effect of climate change on future snow conditions, you could also make reference to this recent work: Verfaillie et al. (2018).

**Response:** The relevant reference has been added. Please see Page 2, Line 22.

8) P.2 - L.25: "particularly in lower elevation and mid-latitude ski areas" → "particularly at lower elevations and in mid-latitude ski areas".

**Response:** Corrected as suggested.

9) P.2 - L.25-26: I do not entirely agree with this statement. In some cases, even at mediumto-long term, snowmwaking could prove to be useful for keeping ski areas reliable, compensating for the scarcity of natural snow (Spandre et al., 2018). While stressing very well the environmental impact related to snowmaking installations, this section does not focus enough on the added-value of machine-made snow production as a necessary complement to natural snow for ski-resorts. In this regard, you could see for example Steiger (2010), Hanzer et al. (2014) and Damm et al. (2014, 2017).

**Response:** The Reviewer is correct. The relevant description has been modified as follows (Page 2, Line 25-29):

"Snowmaking is usually preferred as a supplementary strategy for individual ski areas in response to worsening natural snow conditions (Hennessy et al., 2008; Pons-pons et al., 2012). Machine-made snow can remarkably improve the reliability of snow conditions in both current and future climate scenarios, especially in regions with less reliable snow conditions and short ski seasons (Steiger, 2010; Hanzer et al. 2014; Damm, Koeberl, and Prettenthaler, 2014; Damm et al. 2017)."

10) P.2 - L.30-31: These figures are a bit low. One ha of slope, covered with 30 cm of artificial snow having a density of 500 kg m$^{-3}$, will necessitate a water consumption of 1.5 million litres. However, snowmaking efficiency in converting water into snow is less than 80%. This is mainly determined by wind conditions, according to, for instance, Spandre et al. (2017). Therefore, in practice at least 2 million liters of water are needed to cover 1 ha with a 30 cm thick snowpack. As for the electric consumption, it is usually accepted that the current snow-gun generation requires 2 to 3 kWh to produce 1 m$^3$ of snow.

**Response:** We agree with the Reviewer. These figures have been updated in the revised manuscript, and the text has been modified accordingly (Page 2, Line 31 to Page 3, Line 1):

"The efficiency of converting water into snow during snowmaking is less than 80% because of the influence of meteorological conditions (Spandre et al., 2017). For example, at least 2 million liters of water are needed to cover 1 ha with a 30 cm thick snowpack. In terms of electric consumption, the operation of a snow gun currently requires 2 to 3 kWh to produce 1 m$^3$ of snowpack."

11) P.2 - L.34: "became critical since inappropriate selections" → "is critical since inappropriate choices".

**Response:** Corrected.

12) P.3 - L.1: "To evaluate" → "Evaluating".

**Response:** We used "Evaluating" instead of "To evaluate".

13) P.3 - L.5: An important reference that could be used to provide more information on the state of the Chinese ski market is Vanat (2019), in particular for the number of skier visits. This recent book updates some figures of the previous "China ski industry white book 2017".

**Response:** We appreciate the Reviewer for this suggestion. The references have been updated in the revised manuscript, and we rephrased the following sentences:

"Recently, in major skiing countries, the number of ski resorts has become relatively saturated or declined (such as in Japan, Vanat, 2019)." (Page 3, Line 6-7)

"Thirty-nine new ski areas were opened in 2018, bringing the total number of ski areas in China to 742 (Vanat, 2019), and this is only the start of the upcoming extensive growth in ski tourism." (Page 3, Line 13-14)

"Although there were indeed 742 ski areas in China according to the International Report on Snow and Mountain Tourism in 2018 (Vanat, 2019), only 598 ski areas' information was manually collected from an online map (https://maps.baidu.com), because limited information was available online for lower-class ski areas." (Page 8, Line 23-25)

14) P.3 - L.6: "public to" → "public in China to".

**Response:** We replaced "public to" with "public in China to".

15) P.3 - L.25: "Systems" → "System".

**Response:** Corrected.

16) P.3 - L.26: "linear weighted" → "a linear weighted".

**Response:** Corrected as suggested.

17) P.3 - L.29: In the introduction, it is worth mentioning the work of Demiroglu et al. (2019) and Demiroglu (2019), who have defined a Ski Climate Index (SCI) to estimate the overall suitability of Turkey for ski tourism. This index has been designed as a combination of several factors, among which snow reliability, land cover, aesthetics, market accessibility, comfort (sunshine, wind and temperature conditions) and even security. There are several similarities between your approach and Demiroglu's.

**Response:** We appreciate the Reviewer for providing the information about the work of Demiroglu et al. (2019) and Demiroglu (2019). However, since this paper has not been published, we will follow up this interesting work.

18) P.3 - L.30-33: I would rephrase this section to improve clarity: "This paper is organized as follows: in Sect. 2, we provide a description of the data and method. Sect. 3 presents the results. Section 4 evaluates the method, discuss its limitations and proposes suggestions for the development strategy. The final section contains a brief conclusion and discusses future work."

**Response:** The text has been modified accordingly.

19) P.4 - L.13: Please consider adding a table listing the values of each natural and socioeconomic index, along with their normalization (this has been done only for air temperature, in Table 2). Having a complete table summarizing all the indexes will greatly improve the readability of this section.

**Response:** The Reviewer makes a good point. We present a table listing the information of all parameters of indexes. However, we did not add this table in to the revised manuscript because the less information has been provided by processed socioeconomic indexes. Correspondingly, we added some more explanations in the revision:

**Snow cover**:

"An SCD larger than 100 days and SD greater than 30 cm were taken as the optimal conditions for normalization." (Page 5, Line 7-8)

**Topographic conditions:**

"All pixels with values greater than 30° were assigned a value of 30°. The topographic conditions index was produced by a normalized slope gradient (Fig. 1d). The normalized value was 1 if the slope gradient was between 10° and 20°. The greater the absolute difference is from this interval, the lower the normalization value. When the slope gradient is 0° or larger than 30°, the normalized value is 0." (Page 6, Line 6-10)

**Vegetation:**

"NDVI greater than 0.6 was the optimal value for normalization. The lower the NDVI is, the lower the normalized value." (Page 6, Line 29-30)

**Economic conditions:**

"The higher the nuclear density value is, the higher the normalized value." (Page 7, Line 22-23)

**All indexes processed by cost distance:**

"The higher the cost distance is, the lower the normalized value." (Page 6, Line 19-20)

**Table. The minimum and maximum of parameters for normalization.**

| Indexes | Parameters | Values | | Normalization | |
|---|---|---|---|---|---|
| | | Minimum | Maximum | 0 | 1 |
| Snow cover | Snow cover days | 0 | 151 | 0 | $\geq 100$ |
| | Snow depth (cm) | 0 | 61 | 0 | $\geq 30$ |
| Air temperature | Scores of air temperature | 0 | 1057* | 0 | 1057 |
| Topographic conditions | Slope gradient (°) | 0 | 89 | 0 or $\geq 30$ | 10-20 |
| Water resources | Cost distance to river | 0 | 325262 | 325262 | 0 |
| | Cost distance to lake | 0 | 4823160 | 4823160 | 0 |
| Vegetation | NDVI | 0 | 0.92 | 0.6 | 0 |
| Economic conditions | Kernel density of nighttime light | 0 | 4859210 | 4859210 | 0 |
| | Cost distance to provincial capital | 0 | 7467290 | 7467290 | 0 |
| Distance to a city | Cost distance to city with airport | 0 | 4859210 | 4859210 | 0 |
| | Cost distance to city without airport | 0 | 5540950 | 5540950 | 0 |
| Accessibility of transportation | Cost distance to road | 0 | 2062310 | 2062310 | 0 |
| Distance to a tourist attraction | Cost distance to a tourist attraction | 0 | 1815070 | 1815070 | 0 |

**\*** This figure is obtained by referring to Table 2.

20) P.4 - L.17: "both as supplement and attractions". This is not clear, please rephrase.

**Response:** We deleted this sentence in the revised manuscript.

21) P.4 - L.23: "rule as the most" → "rule and is the most".

**Response:** We replaced "rule as the most" with "rule and is the most".

22) P.5 - L.3: This entire "Snow cover" section should be revisited, since several vital points are quite obscure. What is the period considered to define the winter season? What is the minimum snow depth used to define a snow cover day? What do you mean when saying "snow depth is only taken as a reference for the index of snow cover"? When applying the 100-day-rule, did you consider consecutive or non-consecutive days? What MODIS bands and what products did you use to retrieve the snow cover fraction? How were the SD and SCD indexes normalized? You should also highlight the fact that these observations are for natural snow only and the possibility to add some machine-made snow is taken into account by the next index, that for air temperature.

**Response:** Agreed. According to the Reviewer's suggestion, the section of **Snow cover** has been modified as follows:

(1) In this study, the ski season was defined as the period from November 1 to March 31 (Page 4, Line 4-6):

"Regions where the elevation is higher than 4000 m are unsuitable for skiing due to the lack of oxygen, as are the areas where the maximum air temperature is higher than 10 ℃ for more than 90 days during the ski season (November 1 to March 31)."

For clarity, we changed "ski season" to "skiable days" in the following sentence (Page 4, Line 18-19):

"Tervo (2008) analyzed the viability of nature-based winter tourism enterprises and declared that 90–120 skiable days are adequate for making a profit."

(2) The snow cover day (SCD) is acquired from optical remote sensing snow cover product (the moderate-resolution imaging spectroradiometer, MODIS) rather than from snow depth images. In the revised manuscript, we rephrased the following sentence (Page 4, Line 29-31):

"With a spatial resolution of 500 m, version 4 of the moderate-resolution imaging spectroradiometer (MODIS) snow cover products MOD10A1 and MYD10A1 were used in this study to obtain SCD, and these products were downloaded from the National Snow Ice Data Center (NSIDC, https://nsidc.org/)."

(3) The sentence that "snow depth is only taken as a reference for the index of snow cover." have been rephrased as follows (Page 4, Line 22-25):

"However, the snow depth in China is much lower than that in North America and Europe, and the areas with natural snow depths greater than 30 cm are extremely rare (Mudryk et al., 2015). Therefore, we did not use the indicator of the number of days with a natural snow depth greater than 30 cm. As a supplement for the snow depth on ski runs, the average snow depth (SD) during a ski season was considered in this study."

(4) When applying the 100-day-rule, the non-consecutive days have been considered. We added more explanations as (Page 4, Line 25-28):

"In addition, the number of snow cover days (SCD), which is total number of days (can be discontinuous) with snow cover in an area during the ski season, was considered as an indicator of climate suitability (temperature and precipitation) and a measure of the aesthetic value of a site given that many ski areas have snow cover on only the ski runs."

(5) The moderate-resolution imaging spectroradiometer (MODIS) snow cover products (MOD10A1 and MYD10A1, version 4) have been used in this study. The Normalized Difference Snow Index (NDSI) calculated by using the fourth and sixth bands was adopted in MODIS snow cover images to distinguish snow. We revised the sentences (Page 4, Line 29 to Page 5, Line 2):

"With a spatial resolution of 500 m, version 4 of the moderate-resolution imaging spectroradiometer (MODIS) snow cover products MOD10A1 and MYD10A1 were used in this study to obtain SCD, and these products were downloaded from the National Snow Ice Data Center (NSIDC, https://nsidc.org/). The normalized difference snow index (NDSI) was adopted to distinguish snow in the MODIS snow cover images. The NDSI was calculated using the fourth and sixth bands of MODIS, and a pixel was defined as snow if the NDSI value was greater than 0.4. Further details on the MODIS snow cover products can be found in Hall et al. (2002)."

(6) As for the normalization of SD and SCD indexes, we added explanations as (Page 5, Line 7-8):

"An SCD larger than 100 days and SD greater than 30 cm were taken as the optimal conditions for normalization."

(7) As suggested by the Reviewer, we added a short statement to the end of this section (Page 5, Line 9-10):

"This index was for natural snow only, and machine-made snow was taken into account by the index of air temperature."

23) P.5 - L.17: The clarity of this section needs to be improved. First, it is not clear how you have computed daily mean and maximum air temperatures starting from a dataset of daily observations. Second, you do not explain what period you have considered to rule out high-temperature regions and if you require the 90 days with temperature greater than 10°C to be consecutive. Third, you should clearly distinguish between air temperature and wet bulb temperature. The latter is what really matters for snowmaking, and in this case the colder is the temperature, the better is the snowmaking efficiency. For this reason, you should explain that the 11 temperature regimes and their corresponding scores are designed as a trade-off between colder temperatures needed to preserve the snowpack and to produce machine-made snow and warmer temperatures needed by skiers. Finally, the effect of wind, which is not at all considered in this study, should be at least briefly discussed, since it can have a significant impact on both the attractivity of a site and its potential to sustain snowmaking.

**Response:** Points well-taken. We agree that it is important to provide detailed info of the index processing. For clarity, we try to address the issues point-by-point with changes in the revised manuscript.

(1) To explain more clearly how we have computed daily mean and maximum air temperatures starting from a dataset of daily observations, we inserted the following sentences (Page 5, Line 18-21):

"The gridded daily observation dataset includes two variables: daily mean air temperature and daily maximum air temperature. The daily mean air temperature was the average value of the dry-bulb temperature at 02:00, 8:00, 14:00 and 20:00 (local time). The daily maximum air temperature was measured using a maximum thermometer."

(2) The definition of the high-temperature regions has been rephrased as follows (Page 5, Line 24-26):

"In the southern China, many small ski areas can be profitable during a ski season with 60 skiable days. Therefore, the high-temperature region was defined as the area with more than 90 noncontinuous days with maximum air temperatures greater than 10 ℃ during a ski season (151 days)."

(3) The dry-bulb temperature data have been used in this study, and please see the previous reply to **Air temperature** (1). In addition, the relevant explanation that why we did not use the wet-bulb temperature has been added in Sect. 4.3 (Page 15, Line 15-19):

"The second limitation stems from the air temperature data. Subzero wet-bulb temperatures are needed for snowmaking (Hennessy et al., 2008). Unfortunately, meteorological stations do not provide wet-bulb temperature data. As a proxy, the dry-bulb temperature is also widely used in machine-made snow models (Scott, McBoyle, and Mills, 2003; Steiger, 2010). Therefore, in this study, the dry-bulb temperature data were used to generate the index of air temperature, which reflects the snowmaking conditions and tourist comfort."

(4) The Reviewer makes a good suggestion for the temperature regimes. We added the explanation as follows (Page 5, Line 28-30):

"The 11 temperature regimes and their corresponding scores were designed as a trade-off between the cold temperatures needed to preserve the snowpack and to produce machine-made snow and the warm temperatures needed by skiers."

(5) Finally, the effect of wind has been discussed in Sect. 4.3 (Page 15, Line 20-27):

"Another limitation is associated with wind, which has a significant impact on the efficiency of snowmaking and the attractivity of a site (Spandre et al., 2017). Wind speeds higher than 6 m s-1 have a direct physical or mechanical effect on tourists (Freitas, 2005). Daily wind speed data are available from the meteorological stations that are generally located around a city, while most ski areas are built in the mountains far from cities. Hence, the relatively sparse meteorological stations across the study area did not allow for the creation of reliable daily gridded data. In addition, wind is considered one of the most difficult meteorological variables to model due to its dependence on the specific characteristics of any given location, such as topography and surface roughness

(Morales, Lang, and Mattar, 2012). The local micrometeorology at a specific location also cannot be captured using existing reanalysis wind speed data with low resolution."

24) P.5 - L.21-24: Usually, the slope classification goes like this: very easy = 15% = ∼9°; easy = 20% = ∼11°; hard = 30% = ∼17°; very hard = 40% = ∼22°. Therefore, I would not exclude slopes between 10° and 20°. Please comment on that.

**Response:** Agreed. Following the suggestion, we set the slope gradients of 10° to 20° as the best topographic conditions. The figure 1 (d) has also been replotted.

"Accordingly, slope gradients of 10° to 20° were considered to be the best topographic conditions." (Page 6, Line 3-4)

25) P.5 - L.27: What minimum and maximum values did you use to normalize the slope gradient?

**Response:** Thanks. To make it clearer, we added the details in the revised manuscript (Page 6, Line 6-10):

"All pixels with values greater than 30° were assigned a value of 30°. The topographic conditions index was produced by a normalized slope gradient (Fig. 1d). The normalized value was 1 if the slope gradient was between 10° and 20°. The greater the absolute difference is from this interval, the lower the normalization value. When the slope gradient is 0° or larger than 30°, the normalized value is 0."

26) P.5 - L.33: Why did you consider only rivers and not, for example, lakes?

**Response:** Good point. We agree that the lakes should also be considered. In the revised manuscript, we calculated the index of water resources by integrating river and lake data. The description of lake data has been added in Table 1. Correspondingly, we rephrased the following sentences (Page 6, Line 16-19):

"Data on the spatial distribution of rivers and lakes were acquired from the Data Center for Resources and Environmental Sciences (RESDC, http://www.resdc.cn) of the Chinese Academy of Sciences (CAS). The water resources index is the sum of the cost distance to a river and the cost distance to a lake, each normalized to a range of [0-1] and given a weight of 0.5 (Fig. 1e)."

27) P.6 - L.2: Could you give a few more details on the cost distance method you allude to?

**Response:** We thank the Reviewer for this suggestion. We added the associated descriptions in the revised manuscript (Page 6, Line 20-23):

"The cost distance to a river (lake) was estimated using the cost distance method, which calculated the least cumulative cost distance for each pixel to the nearest river (lake) over a cost surface. The cost surface was a raster of slope gradients, which defined the cost to move planimetrically through each pixel. The value of the slope gradient at each pixel location represented the cost per unit distance to move through the pixel."

28) P.6 - L.5: "Vegetation is a representative of an ideal environment and an important" → "Vegetation contributes to creating an ideal environment and is an important".

**Response:** Corrected as suggested.

29) P.6 - L.7: About wind conditions, please see comment on P.5 - L.17.

**Response:** Please see our previous reply to comment #23.

30) P.6 - L.14: Please explain briefly how the "geometrical interval classification" has been applied in this case.

**Response:** To explain the geometrical interval classification, we inserted the following sentences into the revised manuscript (Page 7, Line 4-6):

"The classification scheme creates 10 geometrical intervals by minimizing the square sum of elements per class. This procedure ensured that each class range had approximately the same number of values for each class and that the change between intervals was fairly consistent."

31) P.6 - L.28: "relative values". Relative to what?

**Response:** We deleted the "relative values". This sentence has been rephrased as follows (Page 7, Line 19-21):

"In this study, without a direct estimation of the GDP values, nighttime light pixel values were used as a surrogate index to measure the economic development level in different areas."

32) P.6 - L.28: "without an estimation" → "without a direct estimation".

**Response:** We replaced "without an estimation" with "without a direct estimation".

33) P.6 - L.30: Please explain briefly how the "kernel density analysis" has been applied in this case.

**Response:** We added the explanations in the revised manuscript as follows (Page 7, Line 23-25):

"All pixels greater than 0 in the composite images of annual average radiance were converted to point features. The kernel density method was used to calculate the magnitude per unit area from the point feature using a kernel function to fit a smoothly tapered surface to each point."

34) P.7 - L.2: Please explain briefly how the "distance decay theory" has been applied in this case.

**Response:** Thanks. This sentence has been revised to provide this information (Page 7, Line 32-33):

"Here, the index of the distance to a city was used to represent the rule of the distance decay theory (the farther away from a city, the fewer visitors the ski area can attract)."

35) P.7 - L.4: Did you apply any selection criterion regarding the size of the cities?

**Response:** Good point. The cities were classified into three categories. We have added the associated descriptions to the section of **Distance to a city** (Page 7, Line 33 to Page 8, Line 5):

"The cities used for this study included provincial capitals, prefecture-level cities and county-level cities with airports. According to the distribution of public-use airports, these cities were divided into three categories: provincial capitals, cities with an airport, and cities without an airport. The distance to a city was estimated by calculating the least cost distance between any point in the study area and the nearest city. The distance to a city index was defined as the sum of the distance to a provincial capital, the distance to a city with an airport and the distance to a city without an airport, each normalized to a range of [0-1] and assigned weights of 0.5, 0.3 and 0.2, respectively (Fig. 2b)."

36) P.7 - L.12: Would it be possible to consider, in addition to the road network, also the distance to airports?

**Response:** We took airport information into account when we classified cities. Please see our previous reply to comment #35.

37) P.7 - L.21: How do you define a ski area? For instance, Vanat (2019), distinguishing between ski areas (designated place where one skis and that may not have lifts) and ski resorts (an organized ski area with more than four lifts), reports a total of 703 Chinese ski areas.

**Response:** Good point. According to Vanat (2019), there are 742 ski areas in China in 2018, and only 149 ski areas have lifts. But we cannot get the information about which ski resorts have lifts. Therefore, in this study we did not distinguish between ski resorts and ski areas (unified use of "ski area").

38) P.7 - L.25: It is not at all clear how the dataset coming from the 128 ski areas has been used to calculate the weight coefficients. See comment on Section 2.3.

**Response:** Please see our subsequent reply to comment #41 (comment on Section 2.3).

39) P.7 - L.27: "five grades, of which 1S to 5S represent" → "five grades from 1S to 5S, representing".

**Response:** Corrected as suggested.

40) P.8 - L.1-3: This sentence is not clear. Do you mean that only 27 areas out of 35 have been ranked, since the others are already listed in the Xinjiang Government Tourist Office? If it is the case, you could rephrase as follows: "According to the feedbacks from our field surveys, ski areas were roughly ranked into different grades, except for the 8 ski areas whose grades were already defined in the Xinjiang Government Tourist Office."

**Response:** The Reviewer is correct. The text has been corrected accordingly (see Page 9, Line 8-9).

41) P.8 - L.4: Section 2.3 needs to be improved in several aspects.

- In Eq.1 and Eq.2, you introduce the weighting factors $W$, while in Eq.11 you compute the weight coefficients $w$. Are they not the same?

- In Eq.1 and Eq.2, $i$ and $j$ are used for the natural and socioeconomic conditions, respectively. Starting from Eq.4, however, the same indexes are used for the number of optional schemes and the number of evaluating indexes. This generates some confusion.

- In Eq.4, you should say what the optional schemes and the evaluating indexes are in your case. If $n$ is equal to 128 and $m$ is equal to 9, this has to be stated clearly.

- In Eq.5, the reason why you have to normalize is not presented clearly.

- In Eq.6 and Eq.7, what does $r$ stand for? Should it not be x instead? And $x_{j,max}$ instead of $x_{max}$?

- In Eq.10, why should $P_{ij}$ be equal to 0?

More generally, the main missing point is the description of how the dataset coming from 128 ski areas has been used to compute $w$, $a$ and $b$. In particular, it seems that the same dataset has been used to compute the coefficients and then to evaluate them. Please clarify this point.

**Response:** We apologize for the confusing description. For clarity, the Sect. 2.3 has been modified accordingly (Page 9, Line 13 to Page 10, Line 23):

"In this study, the linear weighting method was used for synthetic evaluation. The natural suitability for ski area development (NS) is expressed as

$$NS = \sum_{i=1}^{i} W_i NC_i \tag{1}$$

where $i$ is the number of natural condition indexes ($i = 1, 2,…, 5$), $W_i$ is the weighting factor that represents the importance of the natural condition indexes, and $NC_i$ represents the natural index.

The socioeconomic suitability for ski area development ($SS$) is expressed as

$$SS = \sum_{j=1}^{j} W_j SC_j \tag{2}$$

where $j$ is the number of the socioeconomic condition indexes ($j = 1, 2,…, 4$), $W_j$ is the weighting factor that represents the importance of the socioeconomic indexes, and $SC_j$ is the socioeconomic index.

The integrated suitability for ski area development ($IS$) is expressed as

$$IS = \sum_{f=1}^{f} W_f C_f \tag{3}$$

where $f$ is the number of indexes ($f = 1, 2$), $W_f$ is the weight coefficient, $C_f$ is the index of $NS$ or the index of $SS$.

The weight coefficients in Eqs. (1-3) ($W_i$, $W_j$ and $W_f$) are defined using the entropy method introduced as follows. Finally, *NS*, *SS* and *IS* are rescaled within the range of 0-1.

The 116 existing ski areas established before 2012 were used as samples to compute the weight coefficients by extracting all index values for the ski area location. The weight coefficients were calculated by an objective method based on the theory of information entropy, which has been widely employed for the determination of weights of evaluating indicators. (Bian et al., 2018; Bednarik et al., 2010; Srdjevic, Medeiros, and Faria, 2004; Vranešević et al., 2016). The concept of information entropy originally came from thermodynamics and indicates the extent of disorder in the system status (Bednarik et al., 2010). Generally, if the dispersion of data is high, the value of information entropy is low, which means that more information will be provided. Correspondingly, the greater the influence of the index on the evaluation, the higher its weight (Tang, 2014). Therefore, the entropy weight method can be used to calculate the objective weights of the index system and avoid bias caused by subjectivity to a certain extent (Pourghasemi, Mohammady, and Pradhan, 2012).

It is supposed that there are a optional schemes, each with b evaluating indexes. The data matrix is established as follows:

$$X = \begin{bmatrix} x_{11} & \cdots & x_{1a} \\ \vdots & \ddots & \vdots \\ x_{b1} & \cdots & x_{ba} \end{bmatrix}_{a \times b} \tag{4}$$

where $x_{nm}$ is the value of the $m$[th] index in the $n$[th] scheme, $n = 1, 2,…, a$; $m = 1, 2,…, b$. $a$ is the number of ski areas established before 2012 ($n = 116$), $b$ is the number of indexes in Eqs. (1-3) ($i$, $j$ and $f$).

it is noted that the indexes are not measured on the same scale, and they have different dimensions. If the analysis is performed directly with the original index values, the difference in different quantity grades in X may produce inaccurate results in the decision-making process. Therefore, normalization ($Z_{nm}$) is performed to make the indexes dimensionless and mutually comparable, which is expressed as

$$Z_{nm} = \frac{x_{m,max} - x_{nm}}{x_{m,max} - x_{m,min}} \tag{5}$$

where

$$x_{m,max} = max\{x_{1m}, \dots x_{am}\} \tag{6}$$

$$x_{m,min} = min\{x_{1m}, \dots x_{am}\} \tag{7}$$

The information entropy $e_m$ of the $m^{th}$ index is defined as

$$e_m = -k \sum_{n=1}^{a} P_{nm} \ln(P_{nm}) \tag{8}$$

where

$$k = \frac{1}{\ln a} \tag{9}$$

$$P_{nm} = \frac{Z_{nm}}{\sum_{n=1}^{a} Z_{nm}} \tag{10}$$

if $P_{nm} = 0$, $P_{nm} \ln(P_{nm}) = 0$.

The entropy weight $w_m$ of the $m^{th}$ index is expressed as

$$w_m = \frac{1 - e_m}{\sum_{m=1}^{b}(1 - e_m)} \tag{11}"$$

42) P.10 - L.2: "led" → "lead". Sometimes you use the past tense, other times you use the present simple. Please standardize this throughout the text.

**Response:** We thank the Reviewer for pointing out the inconsistency. In the revised manuscript, we have checked the text carefully and unified the verb tenses.

43) P.10 - L.5: "that evaluates" → "by evaluating".

**Response:** Corrected as suggested.

44) P.10 - L.10: "China, which differs" → "China. This patterns differs".

**Response:** Corrected as suggested.

45) P.10 - L.16: You say that the suitability index allows to identify 7 regions, but then you list 8, and in Fig.5 they are 10. Please check this out.

**Response:** We apologize for the inconsistency. We modified the text and Fig. 5 to keep them consistent: "The integrated suitability results identified ten regions that have the greatest potential for ski area development, which were the Changbai Mountains (northeast China), the Daxing'an Mountains (Inner Mongolia Province),the Yanshan Mountains (Beijing-Tianjin-Hebei region), the Lvliang Mountains and the Taihang Mountains (Shanxi Province), the Tianshan Mountains (the Northern Tianshan Mountain Economic Zone), the Qilian Mountains (Qinghai Province and Gansu Province), the Qinling Mountains (Shaanxi Province), the area surrounding Mount Tai (Shandong Province), the Southeast hills (Yangtze River Delta), and the southeastern Tibetan Plateau (Sichuan Province)." (Page 12, Line 5-10)

[Figure]

Figure 5. The spatial distribution of integrated suitability for ski area development based on the natural and socioeconomic indexes. I: Changbai Mountains (northeast China); II: Daxing'an Mountains (Inner Mongolia); III: Yanshan Mountains (Beijing-Tianjin-Hebei region); IV: Lvliang Mountains and Taihang Mountains (Shanxi Province); V: Tianshan Mountains (the Northern Tianshan Mountain Economic Zone); VI: Qilian Mountains (Qinghai Province and Gansu Province); VII: Qinling Mountains (Shaanxi Province); VIII: the area surrounding Mount Tai (Shandong Province); IX: Southeast hills (Yangtze River Delta), X: southeastern Tibetan Plateau (Sichuan Province).

46) P.10 - L.16: "have greatest" → "have the greatest".

**Response:** Corrected.

47) P.10 - L.22-23: Reference to Fig.6 should be made at the end of the previous sentence (after "gradient colors").

**Response:** Corrected as suggested.

48) P.10 - L.24: I would replace "dark-shaded" with "black-shades" and "light-shaded" with "yellow-shaded".

**Response:** The text has been modified accordingly.

49) P.10 - L.31: These surface areas in ha seem to be out by a factor of 10. Please check them again.

**Response:** We apologize for the error. We have updated the figures in the text and Table 4.

"In total, 35.4 million hectares (9.2% of the analyzed area) were categorized as natural-driven areas (Table 4), which were mainly distributed in the mountains of northwestern and northeastern China and the area around the Tibet Plateau. A total of 183.7 million hectares (47.7% of the analyzed area) were classified as socioeconomic-driven areas, including large areas of eastern and central China. A total of 46.1 million hectares (12% of the analyzed area) were determined as ideal areas, particularly distributed

in the Beijing-Tianjin-Hebei region, the eastern part of northeast China and central Xinjiang Province. Finally, 120.2 million hectares of land (31.2% of the analyzed area) were found to be unfavorable areas, which were mainly distributed in regions of northwestern and southern China." (Page 12, Line 21-27)

Table 4. Surface areas and driving factors for four zones of suitability.

| Zones | Natural suitability | Socioeconomic suitability | Area (ha×10⁶) | Percentage (%) |
|---|---|---|---|---|
| Natural-driven areas | 0.5~1 | 0~0.5 | 35.4 | 9.2 |
| Socioeconomic-driven areas | 0~0.5 | 0.5~1 | 183.7 | 47.7 |
| Ideal areas | 0.5~1 | 0.5~1 | 46.1 | 12 |
| Unfavorable areas | 0~0.5 | 0~0.5 | 120.2 | 31.2 |

50) P.11 - L.3: "Moreover" → "Finally".

**Response:** We replaced the "Moreover" with "Finally".

51) P.11 - L.5: Adding up the 4 areas, you get less than half the total area of China. Do the excluded areas account for the rest? Please comment on that.

**Response:** The Reviewer is right. We added the following comment in the revised manuscript as requested (Page 12, Line 27-28):

"The excluded areas (because of high temperature and high elevation) account for more than half of the total area of China."

52) P.11 - L.7: The low suitability here appears to be mostly due to poor natural conditions.

**Response:** We apologize for the unclear expression. We rephrased this sentence as follows (Page 12, Line 29-30):

"Due to the influence from socioeconomic conditions, the integrated suitability was weakened in the areas of the Altai Mountains, the Daxing'an Mountains and the marginal zone of the Tibetan Plateau."

53) P.11 - L.8-9: "In contrast, the integrated suitability of eastern China was enhanced; thus, Shandong province, the Yangtze River Delta and the Beijing-Tianjin-Hebei region were more pronounced (Fig. 5)." → "In contrast, the integrated suitability of eastern China has benefited from better socioeconomic conditions, as in the Shandong province, the Yangtze River Delta and the Beijing-Tianjin-Hebei region (Fig. 5)."

**Response:** Corrected as suggested.

54) P.11 - L.11: "verify the evaluation method" → "evaluate".

**Response:** Thanks. We rephrased the sentence as follows (Page 13, Line 2-3):

"This section first used field survey information to validate the evaluation method, and the method was used to evaluate the locational suitability of ski areas established after 2012."

55) P.11 - L.13: "Verification" → "Evaluation".

**Response:** Thanks. We changed the "Verification" to "Validation and evaluation".

56) P.11 - L.17: "0.5. However" → "0.5, while".

**Response:** Corrected as suggested.

57) P.11 - L.27: "selected" → "can consider".

**Response:** We rephrased the sentence as follows (Page 8, Line 28-29):

"We then considered ski areas that had been in operation for more than 6 years as successful and suitable ski areas."

58) P.11 - L.30: "high" → "medium-to-high".

**Response:** We deleted the sentence.

59) P.12 - L.10: What do you mean by "product homogeneity"?

**Response:** We changed to "similar products".

60) P.12 - L.15: "find" → "found". Verb tenses should be carefully checked to ensure consistency (see comment on P.10 - L.2).

**Response:** We replaced "find" by "found". We apologize again for the inconsistency. In the revised manuscript, we have checked the text carefully and unified the verb tenses.

61) P.12 - L.16: "result" → "results".

**Response:** Corrected.

62) P.12 - L.22: According to Vanat (2019), there are currently about 12 million skiers in China. Is the goal of having 300 million skiers in 3 years realistic? Please comment.

**Response:** The 300 million people include not only skiers but also those who participate directly or indirectly in ice and snow sports, such as skaters and related workers. There are no official statistics on the number of people who participate in winter sports. According to Xinhua news agency, "As of October 2018, more than 800 ski facilities had been built and about 50 million Chinese citizens had skied at least once. In all, about 150 million have participated in winter sports directly or indirectly." The future development of the ice and snow sports industry largely depends on the decision-making and guidance of the government.

63) P.12 - L.24: "Thus, in less developed northwestern China" → "Thus, in this less developed area".

**Response:** Corrected.

64) P.12 - L.31: "number" → "few number".

**Response:** Corrected.

65) P.12 - L.31: See comment on P.12 - L.10.

**Response:** We changed "product homogeneity" to "similar products".

66) P.13 - L.18: In the Conclusion, you could say that this study can pave the way to more detailed and refined analyses based on local data and other sources of information, which represents the next, necessary step to help driving investments in new ski-resorts.

**Response:** We appreciate the Reviewer for this suggestion. The relevant statement has been inserted into the revised manuscript (Page 16, Line 12-14):

"This study can pave the way for more detailed and refined analyses based on local data and other sources of information, which represents the next necessary step to promote investments in new ski areas."

67) P.13 - L.26: "base" → "based".

**Response:** Corrected.

68) P.13 - L.27-28: "the rationality of our suitability evaluation methods was verified based" → "our method to estimate the suitability was evaluated based".

**Response:** The revised manuscript has been corrected accordingly.

69) P.14 - L.2: Please consider adding some references on future snow conditions in China.

**Response:** The Reviewer is correct. We added the related references in the revised manuscript (Page 16, Line 18). The **References** have also been updated.

70) P.14 - L.3: "may become better" → "may increase".

**Response:** Corrected as suggested.

71) P.14 - L.4: "become central" → "central".

**Response:** Corrected.

71) **Figure 1**: In all figures, you used Rainbow palettes, which are known to have several drawbacks (https://betterfigures.org/2015/06/23/picking-a-colour-scale-for-scientific-graphics/). I strongly suggest you to use different color-bars to improve readability.

**Response:** Agreed. We reprocessed all figures that were used Rainbow palettes.

72) **Figure 4**: It looks like there are other regions showing high socioeconomic suitability in Northern and North-Eastern China. Why did you neglect them?

**Response:** The Reviewer is correct. The Northeast Economic Zone has been marked in Fig. 4. In addition, we added relevant descriptions for socioeconomic suitability in northern and northwestern China. Fig. 4 also has been redrawn accordingly:

"Additionally, the Hohhot-Baotou-Yinchuan-Yulin Economic Zone is an important hub connecting northern and western China, and the Northern Tianshan Mountain Economic Zone is the largest economic belt in western China, which is located in the core area of the Belt and Road." (Page 11, Line 31 to Page 12, Line 2)

[Figure]

Figure 4. The spatial distribution of socioeconomic suitability for ski area development based on economic conditions, distance to a city, accessibility of transportation, and distance to a tourist attraction. I: Yangtze River Delta; II: belt encircling the Bohai Sea (including the Beijing-Tianjin-Hebei region); III: Central Plains Economic Zone; IV: Northeast Economic Zone; V: Hohhot-Baotou-Yinchuan-Yulin Economic Zone; VI: Northern Tianshan Mountain Economic Zone.

73) **Figure 6**: You could reverse the y-axis of the color-bar to be consistent with Fig.8. See also comment on P.10 - L.24. Finally, you could maybe swap Fig.6 and Fig.7.

**Response:** Agreed. We modified Fig. 6 accordingly. However, we do not consider it necessary to swap Fig. 6 and Fig. 7, because Fig. 6 shows the result of the aggregation method while Fig. 7 presents the distribution of existing ski areas used for validation and evaluation.

[Figure]

Figure 6. Spatial distribution of the driving factors of ski area development. The natural suitability and socioeconomic suitability values are classified into deciles, generating 100 unique color combinations. Socioeconomic-driven areas are shown in red shades, natural-driven areas in green shades, ideal areas in black shades, and unfavorable areas in yellow shades.

74) **Figure 8**: Please consider using the same symbols as those used in Fig.7. It seems that several ski areas established before 2012 (red triangles) match recent ski areas (blue circles): did you count the same areas twice? You should also remind that natural and socioeconomic suitabilities have different weight coefficients (0.52 and 0.48, respectively), that is why the shaded regions and the green line do not intercept the x and y axes at the same point. Finally, in the label: "less" → "lower", "dotted" → "dashed".

**Response:** Thanks. In the revised manuscript, Fig. 8 have been reprocessed to present a detailed analysis of the locational suitability of the 447 ski areas established after 2012:

[Figure]

Figure 8. The suitability values of existing ski areas established after 2012. (a) Socioeconomic-driven areas, (b) ideal areas, (c) unfavorable areas, (d) natural-driven areas. The deeply shaded region has integrated suitability lower than 1/3 (low suitability), the lightly shaded region has integrated suitability greater than 1/3 and lower than 2/3 (medium suitability), and the white region has integrated suitability greater than 2/3 (high suitability). The green dashed line is the demarcation line with an integrated suitability of 0.5. The shaded regions and the green line do not intercept the x and y axes at the same point because natural suitability and socioeconomic suitability have different weight coefficients (0.52 and 0.48, respectively).

75) **Table 3**: Why is the sum of natural suitability coefficients equal to 1.01, and not 1? You should comment on the fact that the total weight of natural suitability (0.52) is higher than that of socioeconomic suitability (0.48). How was this result obtained? See comment on P.8 - L.4.

**Response:** We apologize for the error caused by rounding. In the revised manuscript, the coefficients in Table 3 have been updated due to the changes of indexes:

Table 3. Weight coefficients for the evaluation indexes.

| Indexes | Weight coefficients |
|---|---|
| Natural suitability | 0.52 |
| Snow cover | 0.36 |
| Air temperature | 0.19 |
| Topographic conditions | 0.32 |
| Water resources | 0.04 |
| Vegetation | 0.09 |
| Socioeconomic suitability | 0.48 |
| Economic conditions | 0.37 |
| Distance to city | 0.29 |
| Accessibility of transportation | 0.18 |
| Distance to a tourist attraction | 0.16 |

In addition, the following sentences have been added to the end of Sect. 2.3 (Page 10, Line 24-26):

"The results show that the total weight of natural suitability (0.52) is higher than that of socioeconomic suitability (0.48), which indicates that natural conditions have a greater impact on the development of ski areas than socioeconomic conditions."

As for the calculation method of weight coefficients, please see our previous reply to comment #41.

76) **Table 4**: "The areas of four zones by different driving factors." → "Surface areas and driving factors for four zones of suitability."

**Response:** Corrected as suggestion.

---

## Author Comment (AC4) · 7 Jun 2019

Please see attached revised manuscript with changes marked in red font.

Please also note the supplement to this comment:
https://www.the-cryosphere-discuss.net/tc-2019-43/tc-2019-43-AC4-supplement.pdf

---

## Author Response (AR3)

Dear Editor,

Many thanks for your acceptation of our manuscript.

A few more minor changes were made in the updated version of our manuscript, and a short list is provided as below. The changes in the revised manuscript are marked in red font.

Page 17, Line 14, we added the data availability.
Page 17, Line 19, we modified the punctuation.
Page 17, Line 22, we modified the punctuation.
Page 17, Line 24, we deleted the sentence relating to financial support.
Page 17, Line 29, we added the financial support.

Best wishes,
Tao Che, on behalf of the co-authors

[revised manuscript text omitted]